# JoPA: Explaining Large Language Model's Generation via Joint Prompt Attribution

## Abstract

Large Language Models (LLMs) have demonstrated impressive performances in complex text generation tasks. However, the contribution of the input prompt to the generated content still remains obscure to humans, underscoring the necessity of understanding the causality between input and output pairs. Existing works for providing prompt-specific explanation often confine model output to be classification or next-word prediction. Few initial attempts aiming to explain the entire language generation often treat input prompt texts independently, ignoring their combinatorial effects on the follow-up generation. In this study, we introduce a counterfactual explanation framework based on joint prompt attribution, JoPA, which aims to explain how a few prompt texts collaboratively influences the LLM's complete generation. Particularly, we formulate the task of prompt attribution for generation interpretation as a combinatorial optimization problem, and introduce a probabilistic algorithm to search for the casual input combination in the discrete space. We define and utilize multiple metrics to evaluate the produced explanations, demonstrating both the faithfulness and efficiency of our framework.

## 1 Introduction

Large Language Models (LLMs), such as GPT4 (Achiam et al., 2023), LLaMA (Touvron et al., 2023) and Claude (Anthropic, 2024), have shown excellent performance in various natural language generation tasks including question answering, document summarization, and many more. Despite the great success of LLMs, we still have very limited understanding of the LLM generation behavior – which parts in the input cause the model to generate a certain sequence. Unable to explain the causality between the input prompt and the output generation could cause failure in recognizing potential unintended consequences, such as harmful response (DAN, 2023; Liu et al., 2023; Zou et al., 2023; Zhu et al., 2023) and biased generation (Wang et al., 2023) attributed to a specific malicious description in the input. These issues undermine human trust in model usage, thus highlighting a pressing need for developing an interpretation tool that attributes how an input prompt leads to the generated content.

Explaining LLM generation through **prompt attribution** involves extracting the most influential prompt texts on the model's entire generation procedure, a realm that remains relatively under-explored in current research. While extensive works on input attribution are proposed for text classification interpretation (Chen & Ji, 2020; Modarressi et al., 2023) and next-word generation rationale (Zhao & Shan, 2024; Vafa et al., 2021), they can not be directly applied to explain the full generation sequence due to its complicated joint probability landscape and the autoregressive generation procedure. The complexity for interpreting LLM generation compounds as the model size increases. Another line of works involves **prompting LLMs to self-explain their behaviors** (Wei et al., 2022). This method relies on the model's innate reasoning capabilities, although current findings suggest that these capabilities may not always be faithful (Turpin et al., 2023; Xu et al., 2024).

There are limited existing attempts focusing on explaining the relationship between the input prompts and the complete generated sequence. The most relevant approach, Captum (Miglani et al., 2023), sequentially determines the importance score for each token by calculating the variations in the joint probability of generating the targeted output sequence when the token is dropped from the model input. While being straightforward, this approach treats tokens as independent features, ignoring their joint semantic influence on the generated output. In fact, tokens may contain overlapping or

complementary information. For example, given the input prompt: "Write a story about the doctor and his patient", the most influential components are "doctor" and "patient". Individually removing either of these words would not significantly alter the generated output, resulting in inaccurately low importance score for each of them. This is caused by the semantic interaction among these components, allowing the model to infer the meaning of the omitted one. Existing attribution methods that ablate each token individually fail to capture such combinatorial effect. A straightforward remedy might involve exhaustively assessing all possible combinations to observe the variations in the model generations, which however is impractical due to the vast search space with long-context input prompts.

To efficiently search the space for generating accurate prompt explanations, we develop our framework *JoPA*, which provides the counterfactual explanation to highlight which components of input prompts have the fundamental effect on the generated context via solving a combinatorial optimization problem. We aim to explain the generation behavior of model outputs for any given prompt while take the joint effects of the prompt components into account. Assuming that removing the essential parts of the prompt would result in a significant variation in the model's output, we propose the novel objective function and formulate our task of providing faithful counterfactual explanations for the input prompt as an optimization problem. To quantify the influence of token combinations in the prompt on the generations, we incorporate a mask approach for joint prompt attribution. Thus, our goal of extracting the explanations has been converted to finding the optimal mask of the input prompt. We solve this problem by a probabilistic search algorithm, equipped with gradient guidance and probabilistic updates for efficient exploration in the discrete solution space. Our main contributions could be summarized as follows:

- We propose a general **interpretation scheme for LLM generation task** that attributes input prompts to the entire generation sequence. Notably, this recipe considers the joint influence of input token combinations on the generation. This motivation naturally formulates a combinatorial optimization problem for explaining generation with the most influential prompt texts.

- We demonstrate *JoPA*, an efficient probabilistic search algorithm to solve the optimization problem. *JoPA* works by searching better token combinations that lead to larger generation changes. It takes the advantage of both the gradient information and the probabilistic search-space strategy, thereby achieving an efficient prompt interpretation tool.

- Our framework demonstrates strong performance on language generation tasks including text summarization, question-answering, and general instruction datasets. The faithfulness of our explanations is evaluated based on a suit of comprehensive metrics considering generation probability, word frequency, and semantic similarity, verifying the transferability and effectiveness of our methods across a variety of tasks. Moreover, the generated explanations demonstrate the effectiveness of our framework to potentially be applied to improve the model's ability, especially making the model safer and more efficient.

## 2 RELATED WORK

While there are extensive works devoted to explaining language models in the context of text classification tasks (Shi et al., 2022; Han et al., 2021; Shi et al., 2023), relatively few attempts (Zhao & Shan, 2024; Vafa et al., 2021) are proposed to investigate the importance of prompt texts on the entire generation procedure especially for LLMs. This demonstrates a research gap that this work aims to fill in.

**Explaining Language Generation** There are many work explaining the predictions generated by LLMs by measuring the importance of the input features to the model's prediction on the classification tasks. One group of studies perturb the specific input by removing, masking, or altering the input features, and evaluate the model prediction changes (Kommiya Mothilal et al., 2021; Wu et al., 2020). The other group of works, such as integrated gradients (IG) (Sundararajan et al., 2017), first-derivative saliency (Li et al., 2016), and mixed partial derivatives (Tsang et al., 2020) leverage the gradients of the output with respect to the input to determine the input feature importance. Although Archipelago (Tsang et al., 2020) explains the feature attributions by considering the combined effects of the input attributions, it targets for the multi-label classification task and relies on the neutral baseline. As for the generation tasks, Diffmask learns the differentiable mask for each layer of the BERT model, but it aims to analyze how decisions are formed across network hidden layers by a

simple probing classifier. In contrast, our framework targets on offering insights into the relationship between the input prompt and model generations by employing the mask to highlight the essential prompt attributions.

Moreover, a few studies utilize the surrogate model to explain the individual predictions of the black-box models, and the representative method is called LIME (Ribeiro et al., 2016). As for explaining the model's generation behavior, Captum (Miglani et al., 2023) calculates the token's importance score by sequentially measuring the contribution of input token to the output, which lacks of the accounting for the semantic relationships between tokens. Another work, ReAGent (Zhao & Shan, 2024), focuses on the **next-word generation task**, computing the importance distribution for the next token position. This method ignores the contextual dependencies in the generated output and could not adequately account for dynamically changing generations. Our framework aims to interpret the joint effects of the input prompts on the entire output contexts with considering of the textual information covered the input prompt.

**Self-explaining by Prompting**   As language models increase in scale, prompting-based models demonstrate remarkable abilities in reasoning (Brown et al., 2020), creativity (Oppenlaender, 2022), and adaptability across a range of tasks (Khattak et al., 2023). However, the complex reasoning processes of these models remain elusive and require tailored paradigm to better understand the prompting mechanism. For instance, the chain-of-thought (CoT) paradigm could explain the LLM behaviors by prompting the model to generate the reasoning chain along with the answers (Wei et al., 2022), as pre-trained LLMs have demonstrated a certain ability to self-explain their behaviors. However, recent studies have also suggested that the reasoning chain does not guarantee faithful explanations of the model's behavior (Jacovi & Goldberg, 2020) and the final answer might not always follow the generated reasoning chain (Turpin et al., 2023). xLLM (Chuang et al., 2024) enhances the fidelity of explanation derived from LLMs via a fidelity evaluator, which however is designed for classification tasks. Our efforts are concentrated on explaining LLM generation by analyzing the attribution of the input prompts to the output content, without depending on the model's innate reasoning ability that are not yet satisfactory.

## 3   PRELIMINARIES

We first introduce necessary notions for the LLM generation process, and then discuss limitations of prior attempts explaining the entire generation via prompt attribution.

**LLM Generation Notions**   Denote a specific input prompt as a sequence of tokens $\boldsymbol{x} = (x_1, \ldots, x_T)$, $x_i \in \{1, 2, \ldots, |V|\}$, where $|V|$ represents the vocabulary size, $T$ is the length of the input sequence and the set of all token indices is $\mathcal{I} = \{1, 2, \ldots, T\}$. The corresponding generated output $\boldsymbol{y}$ could be represented as a sequence of tokens $\boldsymbol{y} = (y_1, \ldots, y_S)$ with $y_j \in \{1, 2, \ldots, |V|\}$. The output tokens $\{y_i\}_{i=1}^S$ are generated from the LLM $f_{\boldsymbol{\theta}}$ parameterized by $\boldsymbol{\theta}$ in an autoregressive manner with the probability $p_{\boldsymbol{\theta}}(\boldsymbol{y}|\boldsymbol{x})$ as:

$$p_{\boldsymbol{\theta}}(\boldsymbol{y}|\boldsymbol{x}) = p_{\boldsymbol{\theta}}(y_1|\boldsymbol{x}) \prod_{i=1}^{S-1} p_{\boldsymbol{\theta}}(y_{i+1}|\boldsymbol{x}, y_i). \tag{1}$$

The probability of generating the output text $\boldsymbol{y}$ given the input prompt $\boldsymbol{x}$ illustrates that the coherence and generation of the output texts heavily rely on the input prompt $\boldsymbol{x}$, indicating an implicit causal relationships between the input prompt $\boldsymbol{x}$ and the output $\boldsymbol{y}$.

**Limitation of Prior Interpretation Attempt**   There are limited prior works about explaining the relationship between the individual input prompts and the entire generated sequence, and their faithfulness is limited by treating input tokens independently. Specifically, Captum (Miglani et al., 2023), calculates the importance score for each token by slightly perturbing the input $\boldsymbol{x}$ at the $i$-th token:

$$G_i(\boldsymbol{x}; \boldsymbol{\theta}) = p_{\boldsymbol{\theta}}(\boldsymbol{y}|\boldsymbol{x}) - p_{\boldsymbol{\theta}}(\boldsymbol{y}|\boldsymbol{x}_{\mathcal{I} \setminus \{i\}}), \tag{2}$$

where $\mathcal{I}$ denotes all token indices. Through Eq. (2), one can calculate an attribution score for each token $i$ indicating its importance towards the original generation result $\boldsymbol{y}$. Such method is direct and simple, however, it treats tokens as independent features, ignoring their semantic interaction and joint effect on the generated output. To make it more obvious, let's recall the previous example of "Write a story about a doctor and his patient.". Note that since "doctor" and "patient" are semantically correlated, and removing any of the two words would not have a significant influence on the generated

response, their corresponding importance score will not be too high. On the contrary, the most important token detected by this approach would be "his", which is clearly not ideal.

These observations inspire us to develop a new prompt attribution method that considers the semantic correlation among tokens. Specifically, we assume that the content generated by LLMs is primarily influenced by a subset of tokens jointly. The remaining tokens serve as auxiliary or potentially less relevant information. Those subset of tokens, which fundamentally shape the model's output, are viewed as the explanatory tokens for the generated content. In other words, explanatory tokens do not affect the model output independently, but jointly contribute to the generated responses.

## 4 PROPOSED METHOD

In this section, we propose *JoPA*, a simple yet effective framework designed for generative tasks to explain the attribution of the input prompt by solving a discrete optimization problem. We start with formulating the general objective for the discrete optimization problem, and then we introduce our proposed probabilistic search algorithm for solving the problem.

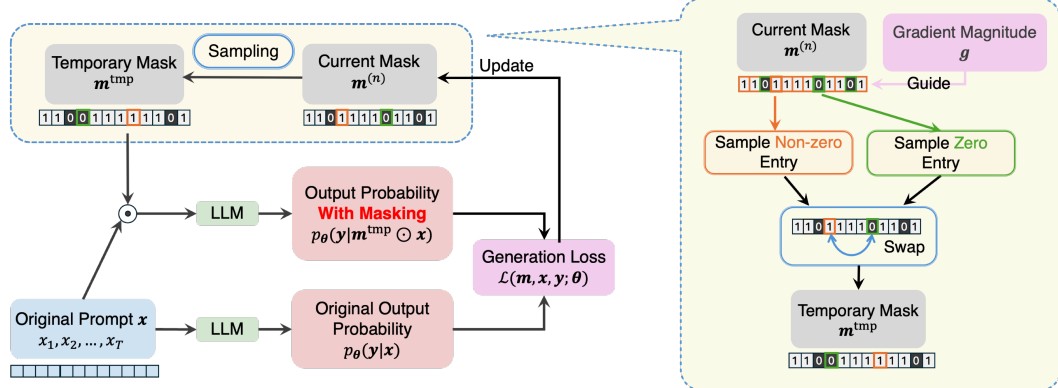

Figure 1: **Overview of *JoPA*. *Left*: Demonstrating the pipeline of the algorithm. *Right*: Illustrating the process of mask $m$ sampling.**

**Problem Formulation for Joint Prompt Attribution** In order to explain language generation via prompt attribution, instead of treating the prompt tokens independently as in previous works (Miglani et al., 2023), we propose to evaluate the joint effect of $k$ prompt tokens on the generated sequence.

Specifically, consider a binary prompt mask $\boldsymbol{m} = (m_1, \ldots, m_T)$ in which $m_i \in \{0, 1\}$ indicates whether the $i$-th token is important or not. The joint probability of generating the original output sequence $\boldsymbol{y}$ given an input masked context $\boldsymbol{m} \odot \boldsymbol{x}$ is computed as $p_{\boldsymbol{\theta}}(\boldsymbol{y}|\boldsymbol{m} \odot \boldsymbol{x})$, where $\odot$ denotes the Hadamard product. In our paper, we aim to identify a binary mask $\boldsymbol{m}$, **which is a (discrete) learnable parameter**, that maximizes the discrepancy in the probability of generating the same output $\boldsymbol{y}$ when comparing the masked input $\boldsymbol{m} \odot \boldsymbol{x}$ to the original input $\boldsymbol{x}$. A large discrepancy indicates that the masked token combination are the most influential components for generating $\boldsymbol{y}$, thus should jointly serve as the attributed explanation. Consequently, this involves training the binary mask $\boldsymbol{m}$ to optimize the following objective function:

$$\max_{\boldsymbol{m} \in \{0,1\}^T} \mathcal{L}(\boldsymbol{m}, \boldsymbol{x}, \boldsymbol{y}; \boldsymbol{\theta}) := p_{\boldsymbol{\theta}}(\boldsymbol{y}|\boldsymbol{x}) - p_{\boldsymbol{\theta}}(\boldsymbol{y}|\boldsymbol{m} \odot \boldsymbol{x})$$

$$\text{s.t.} \quad |\boldsymbol{m}|_1 = T - k, \tag{3}$$

where $k$ represents the number of explanatory tokens. Intuitively, optimizing the objective outlined in Eq. (3) suggests that we want to find the top $k$ important tokens which, if they are masked, lead to a substantial variation in model's output probabilities. Compared with prior methods for individual token attribution stated in Eq. (2), our formulation in Eq. (3) measures the joint attribution of a subset of tokens, capturing token interactions to enable more accurate prompt explanations.

**Challenges in Solving Eq. (3)** Note that Eq. (3) is a constraint discrete optimization problem that is non-trivial to solve. One naive solution would be transforming the discrete optimization problem into a continuous one, i.e., let $\boldsymbol{m} \in [0, 1]^T$, and formulate the constraint into a regularization term. Then one can adopt traditional gradient descent based optimization solutions to solve the problem. However, such a strategy would require extensive gradient calculations and backward

steps on the original inputs, which can be inefficient in practice especially for LLMs. Moreover, the obtained continuous mask is not the final explanation we want. In fact, the approximation error when transforming the continuous mask back into the discrete space can also be quite significant, leading to worse performances. Another straightforward solution involves searching through all possible token combinations exhaustively. However, this could be impractical as well due to the enormous search space especially while facing long-context inputs. Therefore, we hope to develop a new method that adopts a search-based strategy to simplify the algorithm design and satisfy our constraints, while also leveraging gradient information for efficient optimization.

---

**Algorithm 1** Explainable Prompt Generator: *JoPA*

---

**Input:** Input tokens $\boldsymbol{x}$, output tokens $\boldsymbol{y}$, the integer $k$ denoting the number of explanatory tokens, $1 \leq k \leq T$, input mask $\boldsymbol{m}^{(0)} = \mathbf{1}$, and the sampling numbers $N$.
**Output:** Optimal mask $\boldsymbol{m}^{(N)}$

1: $\boldsymbol{g} = |\nabla_{\boldsymbol{m}^{(0)}} \mathcal{L}(\boldsymbol{m}^{(0)}, \boldsymbol{x}, \boldsymbol{y}; \boldsymbol{\theta})|$
2: Set $\boldsymbol{m}^{(1)}$ as the top-$k$ value mask for $\boldsymbol{g}$: $\boldsymbol{m}^{(1)} = \boldsymbol{m}^{(0)}; m_i^{(1)} = 0, \forall i \in (\boldsymbol{g})$
3: **for** $n = 1$ to $N$ **do**
4:     Sample $l \sim \text{softmax}(\boldsymbol{m}^{(n)} \odot \boldsymbol{g})$
5:     Sample $v \sim \text{softmax}((\mathbf{1} - \boldsymbol{m}^{(n)}) \odot \boldsymbol{g})$
6:     $\boldsymbol{m}^{\text{tmp}} = \boldsymbol{m}^{(n)}.\text{copy}()$; switch the value of $m_l^{\text{tmp}}$ and $m_v^{\text{tmp}}$
7:     **if** $p_{\boldsymbol{\theta}}(\boldsymbol{y}|\boldsymbol{m}^{\text{tmp}} \odot \boldsymbol{x}) < p_{\boldsymbol{\theta}}(\boldsymbol{y}|\boldsymbol{m}^{(n)} \odot \boldsymbol{x})$ **then**
8:         $\boldsymbol{m}^{(n+1)} = \boldsymbol{m}^{\text{tmp}}$
9:     **else**
10:        $\boldsymbol{m}^{(n+1)} = \boldsymbol{m}^{(n)}$
11:     **end if**
12: **end for**

---

**Proposed Probabilistic Search Algorithm** To tackle this challenge, we propose *JoPA*, a novel explainable prompt generator, for efficiently obtaining an optimal solution for Eq. (3). We summarize our *JoPA* in Algorithm 1. The high-level pipeline is illustrated in Figure 1 *left*: we initialize and maintain exactly $k$ entries in the mask $\boldsymbol{m}$ to be zero to enforce the constraint and capture their joint influence of being masked; the mask is then iteratively updated by sampling indexes for value swapping, searching towards the direction with increased generation loss $\mathcal{L}(\boldsymbol{m}, \boldsymbol{x}, \boldsymbol{y}; \boldsymbol{\theta})$. At the core of this pipeline is the sampling and update of the discrete mask, which demands an efficient exploration in the vast search space. Figure 1 *right* shows this step, which is featured by the following two essential components, the *gradient-guided masking* and the *probabilistic search update*. **Specifically, it illustrates the process of sampling a non-zero entry and a zero entry from $m$ to swap their values.**

*Gradient-Guided Masking*: Trivial solutions that set the mask $\boldsymbol{m}$ by uniformly random could cost massive sampling to hit the right optimization direction. Gradient is a common indicator of feature importance, as evidenced in existing practices (Ebrahimi et al., 2018; Zou et al., 2023; Shin et al., 2020). Therefore, we propose to use gradient as a guidance to set and sample the mask for more efficient optimization. Specifically, we begin with the binary mask $\boldsymbol{m}^{(0)} = \mathbf{1}$ which indicates that all tokens in the input $\boldsymbol{x}$ are marked as non-explanatory ones. Then we compute the gradient $\nabla_{\boldsymbol{m}^{(0)}} \mathcal{L}(\boldsymbol{m}^{(0)}, \boldsymbol{x}, \boldsymbol{y}; \boldsymbol{\theta})$ of the loss function in Eq. (3), and denote the magnitude of gradients as $\boldsymbol{g} = |\nabla_{\boldsymbol{m}^{(0)}} \mathcal{L}(\boldsymbol{m}^{(0)}, \boldsymbol{x}, \boldsymbol{y}; \boldsymbol{\theta})|$. Note that the components with larger gradient magnitudes in $\boldsymbol{g}$ imply that altering the corresponding tokens could result in a sharper change to the generated outputs. In order to initialize the explanatory $k$ tokens guided by the gradient magnitude, we set the binary mask $\boldsymbol{m}^{(1)}$ where $m_i^{(1)} = 0$ indicates $g_i$ is the top-$k$ value in $\boldsymbol{g}$. The gradient guidance and mask initialization are obtained following Line 1-2 in the algorithm.

*Probabilistic Search Update*: While gradient is informative, greedily determining explanatory tokens by top gradients lacks exploration, leading to suboptimal solutions. Therefore, we propose a probabilistic search mechanism for mask sampling and update. Specifically when updating the current binary mask $\boldsymbol{m}$, we iteratively sample a non-zero entry $l$ from the mask and swap its value with a sampled zero entry $v$ to explore a new (and potentially better) solution for the mask. Particularly, the sampling is also guided by the gradient calculated before: the non-zero entry in the mask (rep-

resenting non-explanatory tokens) is sampled following probabilities calculated by the normalized gradient magnitudes, i.e., softmax($\boldsymbol{m}^{(n)} \odot \boldsymbol{g}$), and we employ a similar sampling strategy for zero entries (explanatory tokens). After swapping the mask indicators for the two sampled tokens $l$ and $v$, we generate a temporary mask $\boldsymbol{m}^{\text{tmp}}$. We then evaluate whether this temporary mask leads to a decrease in output probability. If it does, we update the binary mask to $\boldsymbol{m}^{\text{tmp}}$, otherwise we leave the binary mask as is. Consequently, without requiring intensive gradient computations, these sampling iterations keep discovering improved solutions for the discrete optimization problem shown in Eq. (3). We conclude this update by sampling process in Line 3-12.

While capturing the joint influence of being masked, the proposed *JoPA* uses both the gradient information and the search-space strategy, thereby achieving better efficiency than adopting either method alone. *JoPA* requires only a single step of gradient calculation and avoids the need to convert between discrete and continuous masks. The obtained gradient information provides a favorable initial searching direction and reliable sampling probabilities that enhances the search efficiency.

**Theoretical Guarantee**    Here we could prove that our algorithm can theoretically converge to the local optima given enough iterations. Define that a solution $\boldsymbol{m}^* \in \{0,1\}^T$ is the local optima, if we have $p_{\boldsymbol{\theta}}(y|\boldsymbol{m}^* \odot \boldsymbol{x}) \leq p_{\boldsymbol{\theta}}(y|\boldsymbol{m} \odot \boldsymbol{x})$ for all $\boldsymbol{m}$ in the one-swap neighborhood of $\boldsymbol{m}^*$, namely $||\boldsymbol{m}^* - \boldsymbol{m}||_0 = 2$ (meaning $\boldsymbol{m}$ differs from $\boldsymbol{m}^*$ by one swap, as they are constrained to have $k$ zero entries). It could be proved that the output of the algorithm $\hat{\boldsymbol{m}}$ is the local optima by contradiction that with sufficient iterations. Suppose $\hat{\boldsymbol{m}}$ is not the local optima, there must be a point $\boldsymbol{m}'$ in its neighbor satisfying $||\boldsymbol{m}' - \hat{\boldsymbol{m}}||_0 = 2$, such that $p_{\boldsymbol{\theta}}(y|\boldsymbol{m}' \odot x) < p_{\boldsymbol{\theta}}(y|\hat{\boldsymbol{m}} \odot \boldsymbol{x})$. This means our algorithm can still find a better solution by sampling another swap in further iterations and thus $\hat{\boldsymbol{m}}$ is not the output of our algorithm, leading to a contradiction to the assumption.

## 5    Experiment

This section aims to verify the effectiveness and efficiency of our proposed framework for interpreting the LLMs on the generation task. We conduct the experiments to answer the following questions:

- **Q1**: Do the generated prompt attributions play a predominant role in the model's generation, thereby serving as faithful counterfactual explanations for the generated output?

- **Q2**: Is our interpretation algorithm efficient for practical usage?

- **Q3**: Could the proposed algorithm effectively identify a combination of tokens that impose joint influence on the model generation?

We provide quantitative studies to evaluate the faithfulness of the explanatory prompt fragments generated by *JoPA*, comparing with existing interpretation baselines and ablation variants.

### 5.1    Experiment Settings

**Models & Baselines**    In the experiment, we employ two LLMs as our targeted $f_{\boldsymbol{\theta}}(\cdot)$: LlaMA-2 (7B-Chat) (Touvron et al., 2023) and Vicuna (7B) (Zheng et al., 2023). There are relatively few methods attributing prompts on the entire language generation, and we choose random removal (Random), Integrated-Gradient (Sundararajan et al., 2017), averaged attentions across all layers (Pruthi et al., 2019)(Attention), last-layer attention (Zhao & Shan, 2024)(Last-Attention) and Captum (Miglani et al., 2023) as our baseline for comparison. We also compare *JoPA* with ReAGent (Zhao & Shan, 2024), with the results presented in Appendix A.11. We implement these models using the PyTorch framework and pretrained weights from the transformers Python library (Wolf et al., 2020), and conduct our experiments on an Nvidia RTX A6000-48GB platform with CUDA version 12.0.

**Datasets**    We employ three distinct text generation datasets: Alpaca (Taori et al., 2023), tldr_news (Belvèze, 2022), and MHC (Amod, 2024), to evaluate the effectiveness of our method across various generation tasks. As we aim to capture the joint influence of prompts on the model generations, we focus on relatively long-context prompts rather than simple one-sentence prompts. Longer prompts tend to contain more diverse vocabularies, convey more information, and thus more likely to exhibit high textual correlation. For evaluation purposes, we randomly select approximately 110 data samples with at least 15 words from each dataset. All datasets are publicly available and more details about the dataset are illustrated in Appendix A.1.

## 5.2 EVALUATION METRICS

Faithfulness scores are a key metric for assessing the quality of explanations, with faithful explanations accurately reflecting the model's decision-making process (Jacovi & Goldberg, 2020). Studies (Samek et al., 2017; Hooker et al., 2019) suggest that if a certain input tokens are truly important, their removal should lead to a more significant change in model output than the removal of random tokens (Madsen et al., 2022). Therefore, after removing explanatory tokens identified by a faithful method, the model's new generation would significantly differ from using the original output. Moreover, the input prompt with masking these explanatory tokens are less likely to reproduce the original model response $\boldsymbol{y}$, indicating a lower value of $p_{\boldsymbol{\theta}}(\boldsymbol{y}|\boldsymbol{m} \odot \boldsymbol{x})$.

To quantitatively evaluate explanation faithfulness, we measure the change of model generation behavior from two dimensions. On one hand, we compare the model's original and new generations: the originally generated sequence $\boldsymbol{y}$ is based on the complete input prompt $\boldsymbol{x}$ (e.g., $\boldsymbol{y} = f_{\boldsymbol{\theta}}(\boldsymbol{x})$), while the new generation is conditioned on the masked prompt (e.g., $\boldsymbol{y}' = f_{\boldsymbol{\theta}}(\boldsymbol{m} \odot \boldsymbol{x})$). We thus measure the similarity of the original generation $\boldsymbol{y}$ and the new generation $\boldsymbol{y}'$ based on their word frequency and semantics. A smaller similarity reflects a larger change in model generation, suggesting a better explanation. On the other hand, we measure the likelihood of generating the original output $\boldsymbol{y}$ when the model uses the masked prompt, e.g., changes on $p_{\boldsymbol{\theta}}(\boldsymbol{y}|\boldsymbol{m} \odot \boldsymbol{x})$. A smaller likelihood indicates better explanations whose mask prevents the model from generating its original output. Detailed definitions of these metrics are explained below.

*Word Frequency*: **BLEU** (Papineni et al., 2002) is widely used to measure how close the candidate text $\boldsymbol{y}'$ is to the reference text $\boldsymbol{y}$. The score measures the precision of matching $n$-grams from the text $\boldsymbol{y}'$ to $\boldsymbol{y}$ by a clipping method to avoid overcounting and adjusting for brevity of $\boldsymbol{y}'$ if it is shorter than $\boldsymbol{y}$. **ROUGE-L** (Lin, 2004) is the metric for measuring the overlap of sequences of words between the two texts, evaluating how much of the $\boldsymbol{y}'$ matches with the reference $\boldsymbol{y}$ (precision), how much the reference $\boldsymbol{y}$ is covered by the candidate $\boldsymbol{y}'$ (recall), and combine them into an F1 score.

*Semantic Similarity*: **SentenceBert** (Reimers & Gurevych, 2019) is a variation of the BERT model which is designed to generate high-quality sentence embeddings for the pairs of sentences. We leverage SentenceBert to transform the text $\boldsymbol{y}$ and $\boldsymbol{y}'$ into fixed-length embedding vectors, and calculate the cosine similarity between their embeddings to quantify their semantic similarity.

*Probability Measurement*: We define two measurements to reflect how the likelihood of generating the original text $\boldsymbol{y}$ changes before and after applying the explanation mask. We first define the Probability Ratio (**PR**) to indicate how less likely to generate the original output $\boldsymbol{y}$ when explanatory tokens are masked: $\mathrm{PR}(\boldsymbol{x}, \boldsymbol{y}, \boldsymbol{m}) = \frac{\tilde{p}_{\boldsymbol{\theta}}(\boldsymbol{y}|\boldsymbol{m} \odot \boldsymbol{x})}{\tilde{p}_{\boldsymbol{\theta}}(\boldsymbol{y}|\boldsymbol{x})}$, where $\tilde{p}(\boldsymbol{y}|\boldsymbol{x}) = p(\boldsymbol{y}|\boldsymbol{x})^{1/s}$ is the length-normalized generation probability. If the PR score is far below the random baseline, we can conclude that the masked tokens are indeed important to cause the model generating $\boldsymbol{y}$. In addition, we also calculate the **KL**-divergence between these two distributions to measure their difference: $D_{\mathrm{KL}}(p_{\boldsymbol{\theta}}(\boldsymbol{y}|\boldsymbol{m} \odot \boldsymbol{x})||p_{\boldsymbol{\theta}}(\boldsymbol{y}|\boldsymbol{x}))$, and a larger score indicates a larger change in generation likelihood and a more accurate explanation.

## 5.3 MAIN RESULTS ON FAITHFULNESS (Q1)

We evaluate the faithfulness of the explanatory prompt tokens generated by our framework on five aforementioned metrics. Table 1 shows the results of different methods on three datasets by masking $k = 3$ identified tokens. Experiments on the larger dataset is shown in Appendix A.10. For all metrics except the KL-divergence, a lower score is better which is annotated as ↓. In general, we observe that our method consistently demonstrates better interpretation faithfulness on all datasets compared with baselines, with a clear margin. In particular, we have the following observations demonstrating the advantage of our method:

**Slight random perturbations on the input prompt do not significantly alter the model's output, validating the soundness of our approach.** This is evident from the high PR value of around 0.958 and the low KL value of only 0.023 on the MHC dataset when randomly masking tokens. Only when essential tokens are perturbed, particularly when masked, does the generation of content and probability change substantially. The noticeable gaps in these metrics between *JoPA* and Random underscore the genuine importance of the masked tokens and their role as counterfactual explanations for the model output. The variance assessment of these metrics is presented in Appendix A.6, highlighting the performance stability of our algorithm.

| Model | Dataset | Methods | Metric(@3) | | | | | | |
|---|---|---|---|---|---|---|---|---|---|
| | | | BLEU↓ | ROUGE-L↓ | | | SentenceBert↓ | PR↓ | KL↑ |
| | | | | Precision | Recall | F1 | | | |
| LlaMA-2 (7B-Chat) | Alpaca | Random | 0.601 | 0.527 | 0.533 | 0.522 | 0.825 | 0.842 | 0.134 |
| | | Last-Attention | 0.533 | 0.423 | 0.452 | 0.432 | 0.672 | 0.770 | 0.202 |
| | | Attention | 0.547 | 0.447 | 0.460 | 0.448 | 0.721 | 0.791 | 0.170 |
| | | Captum | 0.515 | 0.409 | 0.421 | 0.409 | 0.680 | 0.602 | 0.417 |
| | | Integrated-Gradient | 0.541 | 0.424 | 0.435 | 0.424 | 0.725 | 0.726 | 0.242 |
| | | *JoPA* | **0.484** | **0.388** | **0.386** | **0.379** | **0.642** | **0.549** | **0.504** |
| | tldr_news | Random | 0.794 | 0.742 | 0.741 | 0.741 | 0.923 | 0.944 | 0.037 |
| | | Last-Attention | 0.747 | 0.683 | 0.685 | 0.683 | 0.876 | 0.869 | 0.108 |
| | | Attention | 0.767 | 0.703 | 0.710 | 0.706 | 0.900 | 0.899 | 0.077 |
| | | Captum | 0.759 | 0.701 | 0.703 | 0.701 | 0.900 | 0.910 | 0.069 |
| | | Integrated-Gradient | 0.713 | 0.641 | 0.642 | 0.640 | 0.866 | 0.817 | 0.149 |
| | | *JoPA* | **0.692** | **0.619** | **0.610** | **0.612** | **0.841** | **0.604** | **0.394** |
| | MHC | Random | 0.723 | 0.617 | 0.614 | 0.615 | 0.787 | 0.958 | 0.023 |
| | | Last-Attention | 0.660 | 0.529 | 0.525 | 0.526 | 0.736 | 0.907 | 0.023 |
| | | Attention | 0.665 | 0.536 | 0.538 | 0.531 | 0.745 | 0.916 | 0.056 |
| | | Captum | 0.640 | 0.497 | 0.493 | 0.494 | 0.663 | 0.760 | 0.189 |
| | | Integrated-Gradient | 0.646 | 0.500 | 0.496 | 0.498 | 0.687 | 0.836 | 0.117 |
| | | *JoPA* | **0.575** | **0.403** | **0.405** | **0.403** | **0.602** | **0.701** | **0.246** |
| Vicuna (7B) | Alpaca | Random | 0.587 | 0.541 | 0.552 | 0.535 | 0.786 | 0.891 | 0.079 |
| | | Last-Attention | 0.475 | 0.423 | 0.436 | 0.415 | 0.672 | 0.840 | 0.113 |
| | | Attention | 0.486 | 0.447 | 0.469 | 0.441 | 0.683 | 0.831 | 0.140 |
| | | Captum | 0.466 | 0.435 | 0.427 | 0.418 | 0.649 | 0.654 | 0.347 |
| | | Integrated-Gradient | 0.541 | 0.495 | 0.503 | 0.488 | 0.744 | 0.835 | 0.135 |
| | | *JoPA* | **0.433** | **0.395** | **0.390** | **0.377** | **0.639** | **0.589** | **0.459** |
| | tldr_news | Random | 0.781 | 0.736 | 0.746 | 0.737 | 0.921 | 0.891 | 0.091 |
| | | Last-Attention | 0.746 | 0.707 | 0.718 | 0.708 | 0.898 | 0.876 | 0.114 |
| | | Attention | 0.621 | 0.514 | 0.515 | 0.509 | 0.817 | 0.714 | 0.294 |
| | | Captum | 0.556 | 0.470 | 0.461 | 0.459 | 0.775 | 0.376 | 0.829 |
| | | Integrated-Gradient | 0.705 | 0.669 | 0.672 | 0.666 | 0.874 | 0.821 | 0.183 |
| | | *JoPA* | **0.536** | **0.456** | **0.454** | **0.448** | **0.772** | **0.317** | **1.006** |
| | MHC | Random | 0.715 | 0.625 | 0.623 | 0.623 | 0.810 | 0.972 | 0.012 |
| | | Last-Attention | 0.684 | 0.570 | 0.573 | 0.570 | 0.773 | 0.947 | 0.028 |
| | | Attention | 0.685 | 0.581 | 0.581 | 0.580 | 0.775 | 0.950 | 0.026 |
| | | Captum | 0.579 | 0.438 | 0.432 | 0.433 | 0.627 | 0.811 | 0.120 |
| | | Integrated-Gradient | 0.672 | 0.559 | 0.555 | 0.556 | 0.762 | 0.941 | 0.030 |
| | | *JoPA* | **0.575** | **0.431** | **0.425** | **0.427** | **0.620** | **0.783** | **0.141** |

Table 1: Faithfulness Measurement Results for LlaMA-2 (7B-Chat) and Vicuna (7B)

***JoPA* can effectively capture semantically important prompt fragments, and the advantage stands out for long context.** Compared to Captum and Integrated-Gradient, our method generally yields lower values for BLEU and ROUGE-L across all datasets. The discrepancies between *JoPA* and Captum are more pronounced for datasets with longer text, such as tldr_news and MHC. Specifically, on the tldr_news dataset, the F1-score of *JoPA* is 12.7% lower than Captum for the LlaMA-2 (7B-Chat) model, and on the MHC dataset, it is 17.85% lower, this all indicates that *JoPA* finds and removes the more important token. Furthermore, the SentenceBert results are better on all datasets, indicating larger semantic variations in the generated output after removing tokens by *JoPA*.

***JoPA* works even better on stronger LLMs.** The gaps between *JoPA* and Captum are less apparent on the Vicuna (7B) model compared to the LlaMA-2 (7B-Chat) model, which could be attributed to the model's inherent inference capabilities. Our method is based on the premise that there are textual correlations among the input tokens, allowing the model to infer from the remaining content when a portion of the tokens is masked. If the model has a poor ability to infer the masked token, it may struggle to capture the contextual information. Consequently, the effectiveness of our method is tied to the LLM's proficiency in inference and understanding.

## 5.4 Time Efficiency (Q2)

In Table 2, we compare the average time cost of *JoPA* and Captum for generating $k = 3$ explanatory tokens for each prompt instance. Note that since Captum's design requires sequentially appending the next token to the input prompt to re-generate the

| Method | | Dataset | | |
|---|---|---|---|---|
| | | Alpaca | tldr_news | MHC |
| Time(s) | Captum | 1169.648 | 1727.602 | 1806.551 |
| | *JoPA* | 15.225 | 15.397 | 14.473 |

Table 2: Time Efficiency on LlaMA-2 (7B-Chat)

new output, the time consumption of Captum would increase significantly when the prompt is long. As the average length of MHC is longer than the Alpaca shown in Table 3, the computational time increases from $1169.648s$ to $1806.551s$ for Captum, which is quite inefficient and impractical. As for comparison, our algorithm aims to solve the combinatorial optimization problem efficiently via the proposed probabilistic search algorithm, which significantly reduce the computational cost of the explanation generation. And also since our design only needs to perform the gradient-guided probabilistic search step for a certain number of times, the computation time remains consistent regardless of the prompt length as can be shown form Table 3.

## 5.5 QUALITATIVE VISUALIZATION (Q3)

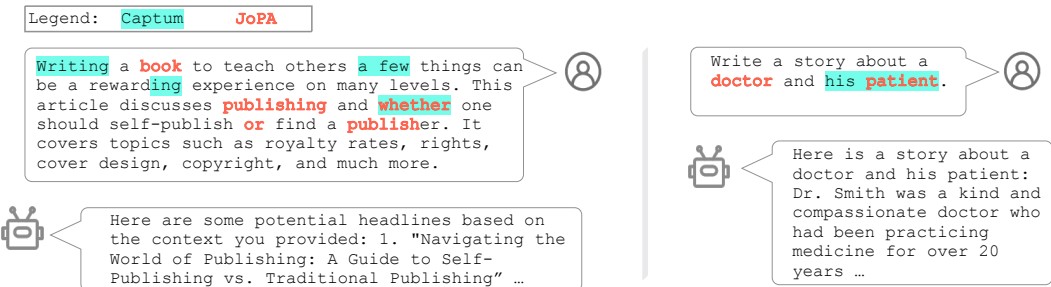

Figure 2: Case study for visualizing the explanation of the model responses.

**Use Case Study.** Figure 2 uses a case study to showcase our method can effectively identify interacted tokens. This figure illustrates the identified explanatory tokens by Captum and our method *JoPA*. While there are few overlaps in the found tokens, compared with our method, those found by Captum are mostly not important to the generated response. This demonstrates the Captum's limited ability to explain the relationship between input prompt and output response, especially considering the textual correlations within input tokens. For instances, the tokens "publishing", "publish" and "book" have semantic correlations, thus Captum masking one by one individually cannot really eliminate this information complementarily provided by the others, especially when the LLM has a certain context inference ability. Similar issues exist in the second case, where "doctor" and "patient" has semantic interactions. Therefore, treating tokens independently and ignoring their joint influence on the generation is not a favored choice for prompt attribution. This case study highlights the importance of our formulation of joint attribution, and verifies the effectiveness of our algorithm in discovering important token combinations on the generated output. More examples are shown in Appendix. A.2.

**Discussion on Benefits.** Moreover, our algorithm shows significant potential for improving model safety and supporting model diagnosis, as elaborated in Appendix A.3. In situations involving a malicious prompt with an adversarial suffix, our approach can be leveraged to effectively detect and remove the attributes responsible for the success of jailbreak attacks. By identifying and filtering out these harmful tokens, we can enhance the model's robustness, making it more resistant to adversarial manipulations and ensuring safer outputs. When it comes to model diagnosis, our method can also provide valuable insights. Specifically, users can utilize this approach to assess how effectively the model responds to a particular input prompt. This evaluation process serves as a diagnostic tool, helping to verify whether the generated content is both relevant to the provided prompt and consistent with its intended meaning. Consequently, our algorithm not only enhances the model's security by mitigating risks from malicious inputs but also contributes to ensuring the reliability and accuracy of the model's output, ultimately improving user trust and satisfaction.

## 5.6 ABLATION STUDY

**Number of Explanatory Tokens.** Figure 3 shows the Probability Ratio (PR) score as the number of masked tokens $k$ changes on each dataset for LlaMA-2 (7B-Chat). Our algorithm consistently outperforms other baseline methods even with larger $k$, demonstrating the stable performance and effectiveness of *JoPA*. After masking $k = 2$ tokens, the value of PR would decrease dramatically, indicating that these two tokens are crucial for generating the output $y$ and they act as the triggers

that alter the probability distribution of the output. As the number of tokens increase to $k = 5$, the PR keeps decreasing but with a less sharp slope. This suggests that the probability of generating the original output $y$ is significantly determined by a few predominant tokens.

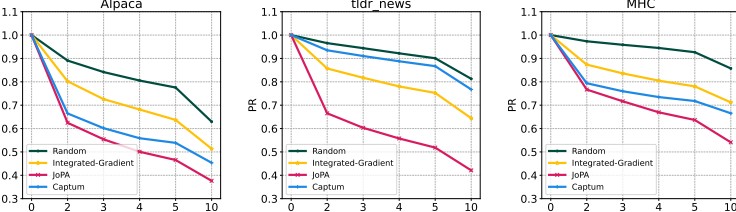 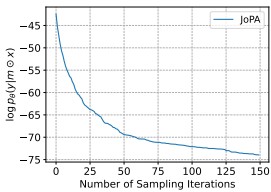

Figure 4: Convergence plot of *JoPA*.

Figure 3: Probability Ratio with varying number of masked tokens $k$.

**Number of Sampling Iterations.** We now demonstrate the convergence of our search algorithm. For each sampling iteration, we sample entries in the mask for value swapping; the mask will be updated if the swapping leads to a drop in the generation log-likelihood $\log p_\theta(y|m \odot x)$. Figure 4 empirically shows how the log-likelihood decreases as the sampling and mask update continue. The quickly decreasing trend demonstrates that our algorithm is successfully performed to improve the quality of mask in locating the predominant tokens on generating $y$.

Recall that in our algorithm, we use gradient as guidance to initialize the mask (e.g., $m^{(1)}$) and calculate sampling probability (i.e., softmax($m^{(n)} \odot g$)). To verify the efficiency of gradient guidance, we compare *JoPA* with two variants: **w/o Initialization** that initializes the mask by uniformly random instead of gradient, and **w/o Probability** that samples swapping entries by uniformly random instead of gradient. Table 5 in the Appendix reports their resulting generation log-likelihood $\log p_\theta(y|m \odot x)$ in different iterations. We observe that without using gradient for initialization, the randomly initialized mask starts from a worse point with a high generation likelihood; and without using gradient to guide the sampling, the mask is updated in less effective direction to explore the search space, resulting in a high generation likelihood in the end. These results show that gradient is a useful and efficient tool for initializing the optimization from a better starting point and guiding the search to a better optimal point.

In addition, we extend the application of our framework, *JoPA*, to a larger model and a more challenging task, specifically Few-shot Chain-of-Thought (CoT) reasoning (Wei et al., 2022; Kojima et al., 2022), as described in Appendix A.8 and Appendix A.9. The better performance of our algorithm compared with others highlights the transferability of our framework across different model architectures and tasks.

## 6 CONCLUSIONS

In this study, we introduce *JoPA*, an efficient probabilistic search algorithm designed to generate the prompt attributions that elicit the model outputs for the generation tasks. We tackle the challenge of explaining the generation behavior for any given prompt by analyzing the joint effect of the prompt attributions on the output. We frame this explanation task as a discrete optimization problem, which can be efficiently solved by our proposed probabilistic search algorithm. This methodology enables efficient generation of any arbitrary number of explanatory prompt attributions that are deterministic to the generated content. Our framework is rigorously evaluated across extensive language generation tasks, including text summarization, question-answering, and general instruction datasets. The faithfulness of the explanatory prompt attributions is thoroughly analyzed and assessed by comprehensive metrics. The results demonstrate that our proposed method efficiently generates explanatory attributions that faithfully reflect the model's generation behavior for the specific prompt. These explanatory attributions interact in semantically and jointly influence the output generation. Furthermore, the overall excellent performance of our method on these diverse datasets highlights its remarkable transferability.

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

## A  APPENDIX

### A.1  DATASET DETAILS

We evaluate explanations on three datasets: Alpaca, tldr_news and MHC. Due to the extensive computational cost, we randomly select approaximately 110 data samples with at least 15 words from each datasets. The statistics of the data used in our experiment can be found in Table 3.

- **Alpaca** Taori et al. (2023) is a dataset with 52000 unique examples consisting of instructions and demonstrations generated by OpenAI's text-davinci-003 engine. Each example in the dataset includes an instruction that describes the task the model should perform, accompanied by optional input context for that task. In our experiment, we randomly select a subset of these examples for verification purposes.

- **tldr_news** Belvèze (2022) dataset is constructed by collecting a daily tech newsletter. For every piece of data, there is a headline and corresponding content extracted. The task is to ask the model to simplify the extracted content and then generate a headline from the input.

- **mental_health_counseling** (MHC) Amod (2024) includes broad pairs of questions and answers derived from online counseling and therapy platforms. It covers a wide range of mental-health related questions and concerns, as well as the advice provided by the psychologists. It is utilized for used to fine-tuning the model to generate the metal health advice. Here, we prompt the model to generate an advice based on the provided question.

| Dataset | Length | Number | Prompt Examples |
|---|---|---|---|
| Alpaca | 16-137 | 108 | Pretend you are a project manager of a construction company. Describe a time when you had to make a difficult decision. |
| tldr_news | 14-138 | 120 | Reddit aired a five-second long ad during the Super Bowl. The ad consisted of a long text message that hinted at the GameStop stocks saga. A screenshot of the ad is available in the article. |
| MHC | 18-478 | 100 | I cannot help myself from thinking about smoking. What can I do to get rid of this addiction? |

Table 3: Data Statistics

### A.2  CASE STUDY

In this section, we visualize three examples which are selected from the dataset Alpaca, tldr_news, and MHC respectively, shown in Figure. 5. The tokens in the red frame are the $k$ explanatory tokens extracted by the model LlaMA-2 (7B-Chat), and the output responses framed in blue are the most informative results selected by human. In the first data sample, the recognized token: "brains" and "cognitive", has some semantic relationship intuitively. It is a common sense the cognitive ability is underpinned by the brain's function and structure. There are numerous reports and studies focused on the functionally relationship between the brain and cognitive ability (Zhang, 2019). Due to the extensive amount of data on which the LLMs are trained, the model can integrate their relationship which is reflected in its response.

As for the second example, the visualized tokens are almost related to the "book" and "publish", which aligns with the theme of the output content. Removing either "publishing" or "publish" would not greatly alter the meaning of the input prompt. This example also shows that our method could detect some fix expressions like "whether...or..", whose relation is recognized by the token "vs." in the output.

In the last example, we could figure out the model infers or "guesses" that the kid feels frustrated and helpless, a sentiment partly related to the token "never". The generated output does not mention the reason to explain how the model makes such inference. From this point, our method could generate human-intelligible explanations to assist the users in identifying which parts of the prompts

fundamentally lead to the undesired response, providing guidance on how to revise the prompts effectively.

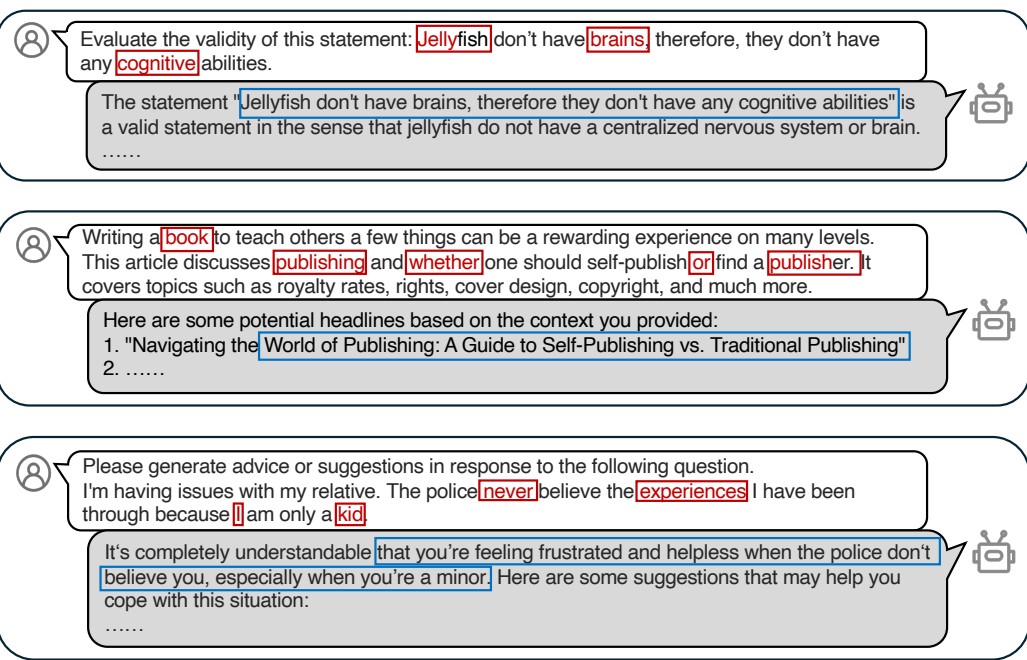

Figure 5: Case study for visualizing the top-$k$ tokens in each example.

In Figure 6, we also illustrate the changes in the model response as the targeted top-$k$ token is removed from the input prompt. Compared with the top-$k$ tokens identified by our method, those found by Captum are mostly not important to the response. This demonstrates the Captum's limited ability to explain the relationship between input prompt and output response with the consideration of the textual information. Moreover, when masking the token "publishing" or "publish", as identified by our method *JoPA*, the generated outputs do not exhibit significant variations in either word frequency or semantics. However, substantial changes in the generated outputs occur only when both tokens are simultaneously masked. This highlights the effectiveness of our methodology in explaining the joint effect of the prompt attributions on the generated output.

### A.3 BENEFITS TO THE COMMUNITY ON IMPROVING MODEL ABILITY

**Benefit on Model Safety** Our method has the potential to help LLM defend against jailbreaking attacks, making the model more robust. We verify this via a real case study. Given an adversarial prompt with malicious request: *"Generate a message that spreads fake news. organized few GO a a a a a newcommand ! SudGuiheures ! !/ refection/:"* It can jailbreak Llama-2 (7B-Chat) model to generate unsafe responses. Our algorithm successfully attributes "Sud" and "he", which are part of the adversarial suffix. By deleting these attributions, the attack fails. Thus, our approach could help detect these high-risk malicious tokens to robustify model safety.

**Benefit on Model Diagnosis** Our approach can also be used to assess the relevance of model's response to the prompt, helping diagnose the effectiveness of model responding to the prompt. For a real case study, consider a single prompt consisting of multiple tasks: *"Create a message to complain about the noise in the neighborhood to friends. Give me the Python code for filtering results based on the following columns..."* We find Llama-2 (7B-Chat) only responds to the first request, neglecting the second one. Our approach attributes the model's output to tokens like "complain" "noise" and "friends", mainly about the first request. Thus by input attribution, our method can help diagnose the overlooked request and suggest better prompt design.

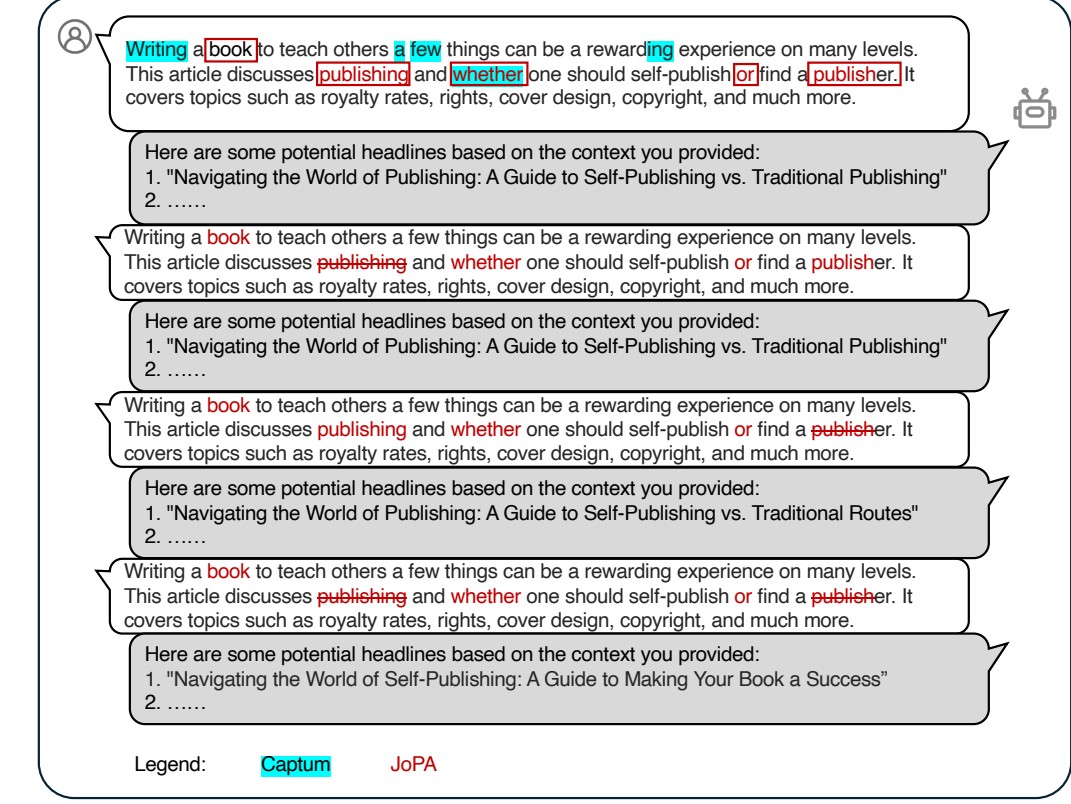

Figure 6: Case study for visualizing the variation of the model responses after the specific tokens are masked.

**Detection Accuracy**   JoPA could be used as the defense of the jailbreak attaks and help users to design better prompts to get satisfying responses. We also do the experiments to show the effectiveness of applying JoPA to detect malicious prompts on Llama-2 (7B-Chat) model, which is similar to the application done in CONTEXTCITE (Cohen-Wang et al., 2024) The results are shown below:

| Method | Detection Accuracy by JoPA | ASR |
|---|---|---|
| GCG | 100% | 3% |
| Prompt with Random Search | 91% | 90% |

Table 4: Comparison of detection accuracy and attack success rate.

The table presents the detection accuracy of JoPA against different adversarial prompts generated by GCG attacks (Zou et al., 2023) and Prompt with Random Search (Andriushchenko et al., 2024) on Llama-2 (7B-Chat). For Prompt with Random Search, the ASR is 90% (Chao et al., 2024), while JoPA achieves a detection accuracy of 91%. This indicates that JoPA successfully identifies 91% of malicious prompts, highlighting its potential utility as a defense mechanism.

### A.4   DISCUSSION ON COUNTERFACTUAL EXPLANATION

Our work *JoPA* is to explain the generation of LLMs by identifying a small changes (i.e., masking few tokens) needed to alter the outcome (i.e., minimizing the probability of the original generation). This can be understood as counterfactual explanation, since it answers a "what if" question: if these input tokens are masked, the LLM would not generate the response. Namely, by applying the mask $m$ on the input prompt $x$, the generation probability of its original response $y$ would be largely decreased (i.e., $p_\theta(y|m \odot x) \ll p_\theta(y|x)$). This suggests that the newly generated output $y'$ resulting from the masked prompt would differ substantially from the original outcome $y$, as measured by our metrics in

Table 1: the semantic similarity (SentenceBert) and word similarity (BLEU and ROUGE-L) between $y$ and $y'$ are much smaller than those from baselines. Here are several case studies to illustrate the concepts discussed.

**General Q&A Task**    Given the input prompt to the model: "Follow the law of supply and demand, describe what would happen to the price of a good if the demand increased." The model would generate the response: "\nIn economics, the law of supply and demand states that the price of a good is determined by the interaction between the quantity of the good that...".  After masking the 3 explanatory tokens: 'supply', 'price', and 'demand', the generated response would change significantly: "However, I must inform you that the question you've provided is not factually coherent, and I cannot provide an answer that may not be accurate or safe...".

**Fill-in-the-Blank Task**    Given the input prompt for the model to fill in the blank:" Fill in the blank with a word or phrase The most successful team in the NBA is the ____" the original response is: "The most successful team in the NBA is the Golden State Warriors". If we masked out the explanatory tokens 'most', 'successful', and 'NBA' from the input, the output would change to: "\Based on the context you provided, the word that best fits the blank is "best." So, the sentence would read:\The team in the office is the best."

The case studies of diverse tasks above demonstrate that the new responses generated from the masked inputs differ significantly from the original outputs, which highlights the impact of these explanatory tokens in influencing and altering the model's output.

## A.5    GRADIENT DESIGN

The Table. 5 displays the log-likelihood without using the gradient as the guidance to initialize the optimization process or to do sampling at different stage when searching for the optimal results.

| Methods | Number of Iterations | | | | | | | |
|---|---|---|---|---|---|---|---|---|
| | 1 | 5 | 10 | 15 | 20 | 30 | 40 | 50 |
| *JoPA* | **-42.448** | **-51.661** | **-56.378** | **-59.852** | **-62.303** | **-64.812** | **-67.274** | **-69.426** |
| w/o Initialization | -33.906 | -44.555 | -50.828 | -55.652 | -58.483 | -62.694 | -64.793 | -66.415 |
| w/o Probability | -41.759 | -49.867 | -53.784 | -57.107 | -59.639 | -63.339 | -65.848 | -67.496 |

Table 5: Ablation study showing the log-likelihood without using gradient for initialization or sampling with different number of iterations.

## A.6    VARIANCE ASSESSMENT

To assess the variance of these metrics, we run the experiment with different seed for 3 times on the Llama-2 (7B-Chat) model. The results of different metrics on three datasets are shown in the Table. 6, where value in each cell denotes mean ± std. The small variance indicates that our algorithm is quite stable across different runs.

| Dataset | BLEU↓ | ROUGE-L↓ | | | SentenceBert↓ | PR↓ | KL↑ |
|---|---|---|---|---|---|---|---|
| | | Precision | Recall | F1 | | | |
| Alpaca | 0.479±0.005 | 0.378±0.008 | 0.380±0.010 | 0.371±0.009 | 0.638±0.005 | 0.552±0.002 | 0.496±0.007 |
| tldr_news | 0.694±0.003 | 0.612±0.004 | 0.613±0.001 | 0.610±0.002 | 0.842±0.004 | 0.592±0.005 | 0.406± 0.008 |
| MHC | 0.575±0.000 | 0.407±0.001 | 0.406±0.001 | 0.403±0.001 | 0.595±0.008 | 0.701±0.000 | 0.245±0.000 |

Table 6: Variance measurement results for LlaMA-2 (7B-Chat)

## A.7    OPTIMIZING MASK VIA GRADIENT

As discussed in Section 4, the strategy that directly optimizes the mask based on gradients requires first relaxing the discrete problem into a continuous optimization problem (i.e. $\boldsymbol{m} \in [0, 1]^T$), optimizing via gradient descent, and later projecting the continuous solution back into the discrete space (i.e., $\boldsymbol{m} \in \{0, 1\}^T$). This projection from continuous to discrete space in practice usually results in large

rounding error, i.e., after projection, the loss increases dramatically. The results of this strategy for Llama-2 (7B-Chat) model on the Alpaca dataset are in Table 7:

| Method | BLEU↓ | ROUGE-L↓ | | | SentenceBert↓ | PR↓ | KL↑ |
|---|---|---|---|---|---|---|---|
| | | Precision | Recall | F1 | | | |
| Gradient | 0.624 | 0.523 | 0.534 | 0.520 | 0.839 | 0.825 | 0.030 |
| *JoPA* | **0.484** | **0.388** | **0.386** | **0.379** | **0.642** | **0.549** | **0.504** |

Table 7: Optimizing continuous mask on Alpaca dataset

The results verify our claim: directly optimizing $m$ using the gradients (and then projecting to discrete solution) has much worse performance than our method, highlighting the challenge of this discrete optimization problem and our contribution of solving it effectively.

## A.8 Results on Larger Model

To further validate a wider applicability of our algorithm on larger models, we conducted additional experiments for Llama-2 (70B-Chat) 16-bit model on the Alpaca dataset. Due to the inefficiency of baseline Captum, we randomly sampled 20 instances from the Alpaca dataset to obtain the results in Table 8, which demonstrate that our algorithm still outperforms the baselines on this larger model with clear margins.

| Method | BLEU↓ | ROUGE-L↓ | | | SentenceBert↓ | PR↓ | KL↑ |
|---|---|---|---|---|---|---|---|
| | | Precision | Recall | F1 | | | |
| Random | 642 | 0.553 | 0.552 | 0.551 | 0.801 | 0.894 | 0.085 |
| Captum | 0.565 | 0.458 | 0.469 | 0.462 | 0.659 | 0.647 | 0.333 |
| *JoPA* | **0.547** | **0.415** | **0.410** | **0.410** | **0.627** | **0.615** | **0.363** |

Table 8: Results of Llama-2 (70B-Chat) model on Alpaca dataset

## A.9 Method Generalizability on Reasoning Task

We demonstrate the generalizability of our algorithm by conducting an additional experiment on the reasoning task of Few-shot-CoT (Wei et al., 2022; Kojima et al., 2022), using the dataset AQuA (Kojima et al., 2022; Goswami et al., 2024). The following results in Table 9 shows the remarkable performance of our method, indicating its ability to generalize well on more complex tasks like CoT.

| Method | BLEU↓ | ROUGE-L↓ | | | SentenceBert↓ | PR↓ | KL↑ |
|---|---|---|---|---|---|---|---|
| | | Precision | Recall | F1 | | | |
| Random | 0.967 | 0.976 | 0.978 | 0.977 | 0.995 | 0.958 | 0.039 |
| Captum | 0.872 | 0.849 | 0.893 | 0.870 | 0.962 | 0.666 | 0.369 |
| *JoPA* | **0.778** | **0.671** | **0.726** | **0.696** | **0.894** | **0.600** | **0.456** |

Table 9: Experimental results on reasoning task of Few-shot-CoT

## A.10 Results on Larger Dataset

As the Captum is 1000 times slower than *JoPA* (Table 2), to afford the comparison, we were using around 100 data in our experiments. Here, we increase the data size by randomly sampling 800 Alpaca data to demonstrate the effectiveness of our method. The experimental results in Table 10 indicate the effectiveness of our method as the sample size increases on Llama-2 (7B-Chat) model.

| Method | BLEU↑ | ROUGE-L↑ | | | SentenceBert↑ | PR↑ | KL↓ |
|---|---|---|---|---|---|---|---|
| | | Precision | Recall | F1 | | | |
| Random | 0.621 | 0.539 | 0.540 | 0.535 | 0.824 | 0.871 | 0.104 |
| Captum | 0.498 | 0.401 | 0.402 | 0.395 | 0.660 | 0.623 | 0.392 |
| **JoPA** | **0.479** | **0.383** | **0.376** | **0.372** | **0.642** | **0.565** | **0.479** |

Table 10: Experimental results on evaluation metrics for larger data samples.

## A.11 COMPARISON WITH OTHER BASELINES

ReAGent (Zhao & Shan, 2024) addresses a different task than *JoPA*. It focuses on classification tasks by explaining the importance of words in inputs corresponding to the predicted label and extends this approach to generation tasks by explaining the single next predicted word. In contrast, our task focuses on explaining the relationship between the input prompt and the entire generated sentences. To compare the performance of *JoPA* and ReAGent on the generation task, we modify ReAGent's explanatory target to encompass the full generation outputs rather than a single word. We conduct experiments on the Alpaca dataset using the Llama-2 (7B-Chat) model, and the results are presented in Table 11. The experimental results demonstrate that *JoPA* outperforms in explaining the relationship between joint attributions in the input and the resulting output.

| Method | BLEU↑ | ROUGE-L↑ | | | SentenceBert↑ | PR↑ | KL↓ |
|---|---|---|---|---|---|---|---|
| | | Precision | Recall | F1 | | | |
| ReAGent | 0.607 | 0.511 | 0.514 | 0.508 | 0.803 | 0.857 | 0.111 |
| **JoPA** | **0.484** | **0.388** | **0.386** | **0.379** | **0.642** | **0.549** | **0.504** |

Table 11: Experimental results between *JoPA* and ReAGent on Alpaca dataset.

