# OpenReview forum: "JoPA: Explaining Large Language Model's Generation via Joint Prompt Attribution"
_ICLR.cc/2025/Conference — ICLR 2025 Conference Withdrawn Submission_

### Official Review · Reviewer_ZoiX · 2024-10-31

**Soundness:** 2
**Presentation:** 3
**Contribution:** 2
**Rating:** 5
**Confidence:** 4

**Summary:**

This paper primarily addresses the task of prompt attribution, which involves identifying the input tokens that have the greatest impact on model output, and proposes an iterative computation method, JoPA, that can simultaneously identify multiple important tokens. Specifically, JoPA begins with an all-ones mask matrix and calculates the gradient produced on the mask matrix by the results with $p(y|m \odot x)$ (using the mask matrix $m$) and $p(y|x)$ (without the mask matrix). Based on the gradient values at different indices of the mask matrix, JoPA selects the Top-k indices with highest values to fill with zero (where k is a hyperparameter of the algorithm). However, the mask matrix generated by this step alone is not sufficiently effective, thus JoPA requires further iterative optimization. More specifically, during an iteration, JoPA samples from the sets of indices where the values are 0 and 1, according to the previously calculated gradient values, swaps the values of these two indices, and then performs model inference. If the result generated by the model deviates more from the original model output than in the previous iteration, the mask matrix from this iteration is retained; otherwise, it is not. Experimentally, the paper uses multiple metrics such as BLEU, ROUGE, and SentenceBert to measure the disparity between texts generated from the masked input versus the original generation across three datasets on two large language models (LLMs).

**Strengths:**

The proposed method has the following advantages in the task of prompt attribution: JoPA can search for combinations of multiple tokens, rather than one by one; benefiting from the guidance of gradient information, JoPA's search efficiency is relatively high.

Experimentally, the paper uses a robust evaluation system with multiple metrics and dimensions to measure the differences in output before and after input masking.

The writing is logical, comprehensive, and considerate of the problem.

**Weaknesses:**

* **Insufficient contribution to the field.** This work may be a commendable improvement for readers focused on this niche area, but for the broader audience of ICLR, which mainly consists of machine learning and deep learning researchers, the gains from reading this paper might not be significant if they do not engage in prompt attribution tasks. A deeper, impactful work in a niche would fully demonstrate the method's significance in specific applications, but this paper only uses a brief text in the main body and two examples in the appendix to describe this, lacking statistical significance and persuasive power. Additionally, while the paper seeks tokens with the greatest impact on output, it does not consider whether the output contains unsafe or toxic content, as these are aspects of the model's behavior, not simply reducible to output differences. For example, the model might generate two outputs with significant differences, both containing some form of toxicity. However, is the method only limited to the prompt attribution task? I think not. Essentially, the paper identifies the most impactful tokens from the current model input, which could directly apply to detecting important tokens, but conversely, another potentially more interesting application could be in prompt compression (retaining words with significant output impact while removing less impactful ones) or prompt optimization (trying to replace impactful words to improve performance). These are directions worth considering by the authors. If effective in more areas, this paper would undoubtedly be more impressive.

* **Theoretical proofs lack persuasiveness.** In the last part of Section 4, the paper uses reductio ad absurdum to theoretically prove that the method will eventually find a local optimum after infinite searches. However, this lacks persuasiveness. First, it does not prove that a local optimum can be found within a limited number of iterations. It merely states that if a better local point is sampled, it will be accepted by the algorithm, which is obvious; second, the mask is sampled based on gradients, and gradients play a very important role in guiding the mask generation process, yet this factor is not considered in the theoretical proof stage, as there is no explicit evidence that gradient information is correct in performing the prompt attribution task. Thus, the proof may have flaws.

* **Some explanations of concepts are unclear.** For example, the author states in the discussion of related work that some studies are only applicable to next-word prediction, yet the scenario applied by this paper, large language models, also involves next-word prediction, without clearly distinguishing the differences between related work and this paper. Additionally, the paper mentions prompt attribution at the beginning without any explanation, which could be quite unfamiliar to readers not from this field.

* **Experiments to demonstrate method effectiveness are still lacking.** Only extracting 110 data samples per dataset seems too few. Additionally, are there simpler, more direct methods, such as writing some rules and then prompting LLMs to determine the most important words in input sentences? How effective is ChatGPT in this task? Lastly, the paper compares random methods and related works from 2017 and 2023, and an early 2024 study (ReAGent) introduced in the paper, but does not perform experimental comparisons.

* The paper's narrative sometimes uses excessive verbiage; consideration should be given to appropriately simplifying the description of the method.

* **Some typos and format errors.**
1. The headers of Tables 1 and 2 are above the tables, which does not comply with the ICLR template.
2. Tables 1, 2, 3, 5, 6, 7, and 8 lack punctuation in their descriptions.
3. The terms Llama2(7B-Chat) and LLaMA-2(7B-Chat) are used inconsistently.
4. Figures 3 and 4 are not aligned properly.
5. Line 743 should replace a punctuation error; there are many such errors throughout the text.
6. In line 118, "in the the generated output" -> "in the generated output".
7. In line 195, "the pipline of" -> "the pipeline of".

**Questions:**

* The first computational step of Algorithm 1 involves calculating the gradient produced on the mask matrix by the results with $p(y|m \odot x)$ and without the mask matrix $p(y|x)$, but at this point, the mask matrix is initialized as an all-ones matrix, meaning it does not change any model input. In this case, $x$ and $m \odot x$ should be the same, so $p(y|x)$ and $p(y|m \odot x)$ should be the same, and if so, how would a loss of 0 produce a gradient?
* The paper uses Figure 1 to explain its method, but the right side of this figure, which uses gradients to guide mask generation, is too vague and needs revision. Additionally, why would gradients occur on the mask? Was the mask implemented as a learnable parameter?
* The K in the paper's method is a hyperparameter, but in real scenarios, we do not know how many K should be set. Could an optimization algorithm consider this parameter? Intuitively, K would have a significant relationship with the length of the input sentence.

---

> ### Author Response · Authors · 2024-11-21
> **Reponse to Reviewer ZoiX -- Part 1**
>
> We thank the reviewer for the valuable suggestions.
>
> >W1. Insufficient contribution to the ML and DL field.
>
> We believe our work considers an important problem of explaining LLMs' entire generation considering the jointly effect of prompt tokens, offering a unique perspective in LLM interpretability. This initial attempt offers a general and efficient interpretation framework to locate important tokens for the whole generation sequence. And we have demonstrated several use cases for future works to use our framework on potential applications.
>
> **Method applications:** Our method has potential usages on improving the model safety and helping in the model diagnosis as discussed in Appendix A.3. By applying our method to detect malicious tokens in the input prompt for jailbreak attacks, the model can refuse to respond to malicious queries, thereby making the attacks ineffective. Moreover, by identifying the important input attributions, the relevance between the input prompt and output generations will be assessed, contributing to the model diagnosis.
>
> **Without considering unsafe input:** If there are unsafe contents in the input, our method can serve as a defense mechanism, enabling the model to refuse responses to malicious prompts, as discussed earlier. Additionally, our work focuses on explaining the joint effects of input prompts on output generations. While it touches upon safety-related applications, addressing safety issues can be a potential use case but is not the primary focus or main topic of our paper.
>
> **Not limited to prompt attribution task:** The primary goal of our work is to provide explanations for the effects of joint tokens in input prompts on output generations, with a particular focus on capturing their semantic interactions. We uniquely formulate this unresolved challenge as an optimization problem and design a specialized algorithm to address it. While we acknowledge the reviewer's observation that our algorithm may have applications in other research areas, this does not diminish its contribution to the field of XAI. Furthermore, it is important to note that prompt compression and prompt optimization are distinct research topics that fall outside the scope of this work.
>
> >W2. Theoretical proofs lack persuasiveness.
>
> **Local optima:** The local optima proof guarantees that our algorithm could reach a local optimum after sufficient iterations, which is useful in showing that the algorithm is well-behaved without the issue of divergence. Therefore though intuitive, the proof ensures a nice property of the algorithm. We agree that having a theoretical guarantee on the convergence rate or global convergence would be nicer, however, we acknowledge that it is very challenging and still an open problem, considering the complicated optimization landscape. To the best of our knowledge, existing works for explaining LLMs have not yet established such theoretical analysis.
>
> **Use of gradient:** The proof is designed to demonstrate that the algorithm converges to the local optimum, regardless of the mechanism used to propose swaps. The specific strategy for generating swaps under the guidance of gradient is orthogonal to the convergence proofs. Moreover, gradients are commonly used in attribution and feature importance tasks, as they capture the sensitivity of the model's predictions with respect to the input perturbations. The use of gradients has demonstrated success in numerous works and aligns with the common pratices.
>
> The effectiveness of our algorithm is demonstrated through its strong performance across all metrics in Table 1 and the convergence plot in Figure 4. In Table 1, JoPA achieves the lowest probability ratio compared to other methods, indicating its ability to significantly reduce the generation probability of the same output by the masked input. Furthermore, as the number of training iterations increases, the generation probability of the output from the modified prompt stabilizes, as shown in Figure 4. These experimental results collectively highlight the effectiveness of our algorithm.

---

> > ### Author Response · Authors · 2024-11-21
> > **Reponse to Reviewer ZoiX -- Part 2**
> >
> > >W3. Some explanations of concepts are unclear.
> >
> > **Next-word prediction:** Existing work on next-word prediction primarily focuses on explaining the generation of the next predicted word by identifying the preceding words that contribute to it. In contrast, our method aims to explain **the generation of the entire output** given the input prompt. The next-word prediction approach is limited and not well-suited for generation tasks. Firstly, in the era of LLMs, the length of generated outputs is often long and unpredictable, making it insufficient to explain the generation of a single word in isolation. Furthermore, solely explaining individual generated words overlooks the semantic interactions between words and sentences, which are crucial for understanding the overall meaning and coherence of the generated text. Our approach addresses these limitations by considering the broader context and relationships within the output.
> >
> > **Prompt attributions:** Attribution refers to the process of determining the contribution of individual tokens in an input prompt to the model's output [4]. It involves identifying which parts of the input are most influential in shaping the generated response, with the aim of providing explanations for the model's behavior or improving its interpretability.
> >
> > >W4. Experiments to demonstrate method effectiveness are still lacking.
> >
> > **Insufficient data:** As the Captum is 1000 times slower than JoPA (Table 2), to afford the comparison, we were using around 100 data in our experiments. Here, we increase the data size by randomly sampling 800 Alpaca data to demonstrate the effectiveness of our method.
> >
> > || BLEU | ROUGE-L (Precision) | ROUGE-L (Recall) | ROUGE-L (F1)| SentenceBert | PR | KL |
> > |:-----:|:-----:|:-----:|:--------:|:------:|:--------:|:--------:|:--------:|
> > |Random| 0.621 | 0.539 | 0.540 | 0.535 | 0.824 | 0.871 | 0.104 |
> > |Captum| 0.498 | 0.401 | 0.402 | 0.395 | 0.660 | 0.623 | 0.392 |
> > |**JoPA**| **0.479** | **0.383** | **0.376** | **0.372** | **0.642** | **0.565** | **0.479** |
> >
> > The experimental results indicate the effectiveness of our method as the sample size increases on Llama-2 (7B-Chat) model.
> >
> > **Prompt LLMs directly:** Prompting LLMs to provide self-explanations for its responses often results in "black-box" explanations, as the underlying reasoning process remains opaque. Additionally, such explanations are prone to hallucinations [9]. Additionally, crafting prompts for explanations requires careful calibration and design, often including well-constructed examples for in-context learning. Without standardized criteria, it cannot be conclusively stated that prompt-based methods are simpler or more effective than our approach. Designing effective and reliable prompts for explanations remains a unfully-explored topic, and we hope to address this challenge in our future work.
> >
> > **Performance of JoPA on ChatGPT:** The input attribution interpretation methods (ours and baselines) mainly focuses on explaining the generation of open-source LLMs, as it requires access to mask gradients for initialization and model logits for optimizing the token masks. Without access to the internal workings of a model, it is challenging to accurately quantify the importance of token attributions. While this line of interpretation works is not designed for black-box models, we believe it can be integrated to online services by the developers to enhance users' trust to these models.
> >
> > **More baselines:** ReAGent addresses a different task than JoPA. It focuses on classification tasks by explaining the importance of words in inputs corresponding to the predicted label and extends this approach to generation tasks by explaining the single next predicted word. In contrast, our task focuses on explaining the relationship between the input prompt and the **entire generated sentences**. To compare the performance of JoPA and ReAGent on the generation task, we modify ReAGent's explanatory target to encompass the full generation outputs rather than a single word. We conduct experiments on the Alpaca dataset using the Llama-2 (7B-Chat) model, and the results are presented below:
> >
> > || BLEU | ROUGE-L (Precision) | ROUGE-L (Recall) | ROUGE-L (F1)| SentenceBert | PR | KL |
> > |:-----:|:-----:|:-----:|:--------:|:------:|:--------:|:--------:|:--------:|
> > |ReAGent| 0.607 | 0.511 | 0.514 | 0.508 | 0.803 | 0.857 | 0.111 |
> > |**JoPA**| **0.484** | **0.388** | **0.386** | **0.379** | **0.642** | **0.549** | **0.504** |
> >
> > The experimental results demonstrate that JoPA outperforms in explaining the relationship between joint attributions in the input and the resulting output.

---

> > > ### Author Response · Authors · 2024-11-21
> > > **Reponse to Reviewer ZoiX -- Part 3**
> > >
> > > **Excessive verbiage:** We understand the importance of concise descriptions, but we believe the current level of detail is necessary to thoroughly explain our method. Simplifying the narrative further could risk omitting critical information required for a complete understanding of our approach. However, we are open to refining specific sections if you have suggestions on where simplifications would enhance clarity without sacrificing precision.
> > >
> > > > W5. Some typos and format errors.
> > >
> > > Thank you for the detailed suggestions in improving our writing. We will address them in the final version of our paper.
> > >
> > > > Q1. How would a loss of 0 produce a gradient?
> > >
> > > We are sorry for the confusion. Please note in the loss function $\mathcal{L}(m, x, y; \theta)$, the first term is a constant to $m$, therefore $\nabla_{m}\mathcal{L}(m, x, y; \theta)$ is essentially calculated by the second term (i.e., -$\nabla_{m} p_{\theta}(y|m\odot x)$), which produces a valid gradient when $m$ starts from all-ones and is contrained to have k zeros.
> > >
> > > > Q2. Figure 1 needs revision. Why gradients occur on the mask?  Is mask implemented as a learnable parameter?
> > >
> > > Figure 1 right is an illustration for Algorithm line 4-6, which basically samples a non-zero entry and a zero entry from $m$ to swap their values. This is to probabilistically search a new solution from a one-swap local space, as explained in Line 243-251. We will include more explanation for Figure 1 and replace it with a higher resolution version. We are open to making adjustments to Figure 1 based on more sepecific suggestions from the reviewer.
> > >
> > > Yes, the mask is a (discrete) learnable parameter, as indicated in Eq. (3). To optimize it, as discussed in Line 211-218, one solution is to relax the binary mask $m$ as a continuous parameter and directly use projected gradient descent, which however has high projection error. Therefore, we propose a search-based optimization algorithm: we first initialize the binary mask as $m^{(0)}=1$, and compute its gradient to guide the probabilistic search (i.e., $g = |\nabla_{m^{(0)}}\mathcal{L}(m^{(0)}, x, y; \theta)|$), which is also discussed in paper Line 252-263.
> > >
> > >
> > > > Q3. Design a new optimization algorithm with adaptive $k$.
> > >
> > > We agree that making $k$ adaptive is an interesting improvement. Here we follow the interpretation setup of previous works [2, 3] by setting a fixed value of $k$ to ensure a fair comparison across all interpretation methods (e.g., with the same mask budget $k$). In practice, users may customize different values of $k$ based on their specific need, and model providers could offer multiple results with varying $k$ values to accommodate user preferences. We believe that adaptively determining the optimal number of important tokens for different semantic contexts remains an open question and can be an important furture work.

---

> ### Author Response · Authors · 2024-11-25
> **Reponse to Reviewer ZoiX -- Follow-up**
>
> Dear Reviewer ZoiX,
>
> We greatly appreciate your initial comments and we totally understand that you may be extremely busy at this time. However, we kindly hope you can take a moment to review our responses to your concerns. We greatly value any feedback you can provide and would appreciate it if you could consider updating your rating if your questions have been adequately addressed. Additionally, we are more than happy to answer any further questions you may have before the discussion concludes.
>
> Best Regards,
>
> Authors of Paper7871

---

> > ### Comment · Reviewer_ZoiX · 2024-11-26
> > **additional comments**
> >
> > Hi, thank you for your response. Some of my concerns have been well addressed. However, I stand by my scores, as this submission still requires significant revision. That said, I like the idea here, and the work itself is interesting.

---

> > > ### Author Response · Authors · 2024-11-26
> > >
> > > Dear Reviewer ZoiX,
> > >
> > > We sincerely appreciate your thoughtful feedback.
> > > In terms of the revision, in fact, we **have revised our submission based on your comments accordingly** (for your convinence, we just highlighted them in red in our paper), so we are not sure what “significant revision” you are referring to. We hope you could take a look at our revision. If you still have concerns, we appreciate if you can directly tell us what specific kind of revision you would like to see and we are more than happy to further address it for you.  Otherwise, we kindly request you to consider increasing the score.
> > >
> > > Best Regards,
> > >
> > > Authors of Paper 7871

---

> > > > ### Comment · Reviewer_ZoiX · 2024-11-26
> > > >
> > > > Thank you for your reminding. I have reviewed the revised manuscript and all other reviewers' comments. Let me explain a bit further. The reason I have not changed my score is that some of my comments would require a rewrite of parts of the paper and additional systematic experiments. For example, the overall contribution of the paper seems modest, and the data selected for the experiments is quite limited. These issues cannot be completely resolved by merely making simple modifications to the current version. I can see that the authors have tried their best to address these issues in a very short time. However, this does not mean that the issues have disappeared. For conferences like ICLR, I still insist on maintaining a high standard. Note that I am not intentionally being harsh. In fact, I also noticed that other reviewers' ratings are more positive. But we must also recognize the existing problems with this submission, which might require more detailed and systematic revisions.

---

### Official Review · Reviewer_AFbs · 2024-11-02

**Soundness:** 3
**Presentation:** 3
**Contribution:** 3
**Rating:** 6
**Confidence:** 3

**Summary:**

The authors identify a weakness of current prompt attribution methods : The interaction effects of tokens within the prompt are usually disregarded because of the combinatorial cost of evaluation.  To enable a search for the 'most important' tokens, they test a mask optimisation procedure where a fixed number $k$ mask positions are chosen for evaluation, and then local moves (one masked position is moved for each new evaluation) are made to search for the best set of mask positions overall.  A number of experiments are performed and the results show that the method produces better sets of attributions than previous methods.

**Strengths:**

The problem that has been identified (i.e. others not looking at the joint effects from prompts) seems real.  There also seems to be a genuine advantage in searching for a better optimum point using the local search method proposed.

**Weaknesses:**

L288 : The "Theoretical Guarantee" proof that the procedure "theoretically converge to the local optima given enough iterations" (L288) seems tautological, since the definition of local implies accessibility by 1-step swaps.  Clearly a global convergence proof would be much nicer to have, but the local version does not seem like it provides any real 'power' in the necessary direction.

L434 : "our algorithm solves the combinatorial optimization problem efficiently" seems like it is claiming global optimisation.  Perhaps "aims to solve"?

Minor tweaks:
* L014 : "elucidating and explaining" ... are these technically different?  Would "understanding" work instead?
* L024 : "both faithfulness and efficiency" -> "both the faithfulness and efficiency"
* L032 : "limited understanding on" -> "limited understanding of"
* L073 : "probabilistic updaten for" -> "probabilistic updates for"
* L139 : "notions depicting the LLM" -> "notation for the LLM"
* L142 : "Notions on LLM Generation" -> "LLM Generation Notation"

**Questions:**

How was the value of $k$ chosen?  And (to get an idea of the scale of the combinatorial problem) how many tokens are generally within a prompt?

In the initialisation of L266 - how often are the final $k$ mask positions chosen from the initial (say) top-$2k$?  Could we safely reject all but the top-$2k$ tokens from consideration?

Suppose we start with two different mask patterns (for a given $k$, say : 0..01.....1 and 1.....10..0).  What percentage of the time would Algorithm 1 arrive at the same final mask for both?  Figure 4 suggests that even a sequence of gradient-proposed local moves moves towards a global minimum slowly.

---

> ### Author Response · Authors · 2024-11-21
> **Reponse to Reviewer AFbs**
>
> We thank the reviewer for the valuable questions and suggestions.
>
> >W1. A global convergence proof would be nicer.
>
> The local optima proof guarantees that our algorithm could reach a local optimum after a sufficient number of iterations, which is useful in showing that the algorithm is well-behaved without the issue of divergence. Moreover, our proof demonstrates the training procedure satisifes the necessary conditions for convergence, maximizing the probability discrepency of generating the same output by different input.
>
> We agree that a thorough theoretical analysis on the global convergency can greatly strengthen our work. However, we ackowledge that it is very challenging and still an open problem, considering the complicated optimization landscape. To the best of our knowledge, existing works for explaining LLMs have not yet established such theoretical analysis. We believe this is a valuable perspective and will leave this opportunity as our future work.
>
> Although we cannot theoretically prove the global convergence of the algorithm, its effectiveness is empirically demonstrated through its strong performance across all metrics in Table 1 and the convergence plot in Figure 4. In Table 1, JoPA achieves the lowest probability ratio compared to other methods, indicating its ability to significantly reduce the generation probability of the same output by the masked input. Furthermore, as the number of training iterations increases, the generation probability of the output from the modified prompt stabilizes, as shown in Figure 4. These experimental results collectively highlight the effectiveness of our algorithm.
>
> >W2. "Aim to solve" rather than "solve" the optimization problem.
>
> Thanks for the suggestion. We will change it to "aim to solve" to avoid possible misunderstanding.
>
>
> >W3. Minor tweaks.
>
> Thank you for the detailed suggestions in improving our writing. We will revise them in our final version.
>
> **Questions:**
>
> >Q1. The value of k; the number of tokens within a prompt.
>
> Table 1 displays the performance of JoPA when $k=3$ (Line 363). We also show the variation of probability ration (PR) with different $k$ in Figure 3. These results demonstrate the effectiveness of JoPA under various $k$. The summary of the user prompt length for different dataset is in Appendix A.1. In addition, we use a system prompt for all datasets: "You are a helpful, respectful and honest assistant. Always answer as helpfully as possible, while being safe. Your answers should not include any harmful, unethical, racist, sexist, toxic, dangerous, or illegal content. Please ensure that your responses are socially unbiased and positive in nature.\n\nIf a question does not make any sense, or is not factually coherent, explain why instead of answering something not correct. If you don\'t know the answer to a question, please don\'t share false information.", which covers 117 tokens.
>
> >Q2. How often are the final mask positions chosen from the initial (say) top-2k? Could we safely reject all but the top-2k tokens from consideration?
>
> We found that 14.8% of the data samples from the Alpaca dataset do not have at least one final mask position that appeared in the initial top-$2k$ positions. As there are some samples whose final masks are not selected from the top-$2k$ tokens, relying solely on the $2k$ tokens obtained from the initialization would not be sufficient to achieve the final optimal mask, which suggests the necessity of exploring other tokens via our probabilistic search mechanism.
>
> >Q3. Suppose we start with two different mask patterns (for a given k, say : 0..01.....1 and 1.....10..0). What percentage of the time would Algorithm 1 arrive at the same final mask for both? Figure 4 suggests that even a sequence of gradient-proposed local moves towards a global minimum slowly.
>
> In order to verify the stability of our algorithm with various inistializations, we compare the log-prob $logp_{\theta}(y|m\odot x)$ of our gradient-guided initialization with the random initialization by using different $k$ across different iterations N.
>
> |||Gradient-based Initialization|Random Initialization|
> |:-:|:-:|:-:|:-:|
> |k=2|N=1|-29.36|-24.06|
> ||K=50 | -52.59 | -50.09 |
> ||N=150|-53.19|-53.93|
> |k=4|N=1|-42.44|-33.90|
> ||N=50| -69.22 | -66.29 |
> ||N=150|-73.97|-73.95|
> |k=10|N=1|-68.50|-58.47|
> ||K=50 | -99.73 | -96.45 |
> ||N=150|-108.32|-107.11|
>
> The results above indicate that regardless of the initializations, the algorithm consistently converges to a similar local optimum after a sufficient number of iterations, highlighting its stability. The gradient-based initialization could provide a better starting point to achieve a faster convergence than the random initialization. Furthermore, our algorithm is computationally efficient. As shown in Table 2, it outperforms Captum in efficiency, requiring only about 15 seconds to identify the optimal mask.

---

> > ### Comment · Reviewer_AFbs · 2024-11-24
> >
> > [Re: W1. A global convergence proof would be nicer.]
> >
> > This is a slight mis-characterisation of the issue.  The local proof
> > says (essentially, please correct this if it is an over-simplification) :
> > "If we define 'local' to mean the area in which we are searching, then we find the local optimum".  The push-back
> > is against math-ification that doesn't really achieve anything.
> >
> >
> > [Re: Q1. The value of k; the number of tokens within a prompt.]
> >
> > The value of $k$ analysis is fine - what was more important was the number of tokens
> > (so a reader can judge how bad a combinatorical problem the optimisation is trying to tackle).
> > But you brought up an interesting issue : Are the $k$ masks allowed to interfere with the System Prompt?
> > How often are tokens in the System Prompt driving decisions?
> >
> >
> > [Re: Q2. How often are the final mask positions chosen from the initial (say) top-2k?]
> >
> > Good to understand this - so it seems like a simple early-optimisation wouldn't be certain of helping much.
> >
> >
> > [Re: Q3. Suppose we start with two different mask patterns ]
> >
> > Again, this was useful to see.
> >
> >
> > [Overall]
> >
> > I've nudged up my Rating (also based on the Authors responses to other Reviewers)

---

> > > ### Author Response · Authors · 2024-11-25
> > >
> > > Thank you for recognizing our work and for increasing our score.
> > >
> > > For the question about the proportion of $k$ in the system prompt, we only mask $k$ tokens in the user prompt and the system prompt is designed to establish guidelines for the assistant's behavior, ensuring its responses are helpful, safe, honest, and unbiased.

---

> > > > ### Comment · Reviewer_AFbs · 2024-11-26
> > > >
> > > > Thank you for answering my question about the system prompt.  It explains why I was puzzled to hear that the system prompt is 117 tokens in length, since the length of the system prompt is immaterial to your method.

---

### Official Review · Reviewer_4Cfs · 2024-11-02

**Soundness:** 3
**Presentation:** 3
**Contribution:** 2
**Rating:** 6
**Confidence:** 4

**Summary:**

This paper tackles the problem of prompt attribution, i.e., identifying the most important tokens in the prompt that influence the language model generation. This is formulated as a search / discrete optimization problem where the objective is to optimize a binary mask over the prompt tokens that would induce the most significant probability drop in generating the original target output.

The proposed optimization method JoPA is (on a high-level) an improved version of MCMC, with two tricks:

1. Gradient guided masking: with the initial masking m_0, sample generations, use the gradient of the loss function to decide m_1 (leveraging the intuition that higher gradient magnitudes indicate more important t tokens).

2. Probabilistic search update: at every iteration, perturb the current masking by swapping a non-zero entry with a sampling zero entry,  where this sampling is also guided by the gradient. Evaluate with this new perturbed masking, accept it if it improves the optimization objective.

**Strengths:**

- The paper is clearly organized and easy to read. The proposed algorithm is described clearly.
- The experiment setup is very reasonable, with comprehensive evaluation metrics.
- The empirical experiment results are positive.

**Weaknesses:**

- I would love to see a couple more baselines to be compared. For example, I think leave-on-out is a classic baseline not included here. You could also compare with attention-based methods (e.g., selecting most important tokens with highest attention weights). I know there are flaws with these methods but there are all well-known baselines for token-level attribution that I think would be good to include.

- I would love for the authors to further highlight the novelty of the proposed method. Right now it feels like a pretty standard discrete optimization method. Gradient-guided search is nothing new and it's been widely used in prior adversarial attack / attribution literature (e.g., "Universal and Transferable Adversarial Attacks on Aligned Language Models"; "HotFlip: White-Box Adversarial Examples for Text Classification"). And the idea of sampling a perturbation and updating the current masking with some acceptance criteria is also standard discrete optimization style (e.g., MCMC).

- This is pretty minor but the case study in Sec 5.5 isn't super informative to me. Maybe you could consider using some downstream applications to illustrate the usefulness of the proposed method, like the experiments done in a recent (concurrent) paper: "CONTEXTCITE: Attributing Model Generation to Context".

**Questions:**

N/A

---

> ### Author Response · Authors · 2024-11-21
> **Reponse to Reviewer 4Cfs -- Part 1**
>
> We thank the reviwer for the constructive suggestions.
>
> >W1. More baselines, e.g., leave-one-out and attention-based methods.
>
> We appreciate the suggestion, and will include the baselines in our experiments.
>
> - **Leave-one-out method**: We have compared with Captum [7] in Table 1, which is essentially a leave-one-out baseline by masking the token sequentially and use the variations in the output generation probability to indicate the importance of the masked token (Line 112-115).
> - **Attention-based methods**: We use the last layer attention and the average of all layer attentions of the  for measuring the token importance and use the top-$k$ tokens as the most important joint attributions.
>
> We extend Table 1 to include the performance of attention-based methods on Llama-2 (7B-Chat) model:
>
> | || BLEU | ROUGE-L (Precision) | ROUGE-L (Recall) | ROUGE-L (F1)| SentenceBert | PR | KL |
> |:--------:|:-----:|:-----:|:-----:|:--------:|:------:|:--------:|:--------:|:--------:|
> | Alpaca|Attn(Last)| 0.533 | 0.423 | 0.452 | 0.432 | 0.672 | 0.770 | 0.202 |
> | |Attn(AVG)| 0.547 | 0.447 | 0.460 | 0.448 | 0.721 | 0.791| 0.170 |
> | |Captum | 0.515 | 0.409 | 0.421 | 0.409 | 0.680 | 0.602 | 0.417 |
> | |**JoPA**| **0.484** | **0.388** | **0.386** | **0.379** | **0.642** | **0.549** | **0.504** |
> | tldr |Attn(Last)| 0.747 | 0.683 | 0.685 | 0.683 | 0.876 | 0.869 | 0.108 |
> | |Attn(AVG)| 0.767 | 0.703 | 0.710 | 0.706 | 0.900 | 0.899 | 0.077 |
> | |Captum | 0.759 | 0.701 | 0.703 | 0.701 | 0.900 | 0.910 | 0.069 |
> | |**JoPA**| **0.692** | **0.619** | **0.610** | **0.612** | **0.841** | **0.604** | **0.394** |
> | MHC |Attn(Last)| 0.660 | 0.529 | 0.525 | 0.526 | 0.736 | 0.907 | 0.023 |
> | |Attn(AVG)| 0.665 | 0.536 | 0.528 | 0.531 | 0.745 | 0.916 | 0.056 |
> | |Captum | 0.640 | 0.497 | 0.493 | 0.494 | 0.663 | 0.760 | 0.189 |
> | |**JoPA**| **0.575** | **0.403** | **0.405** | **0.403** | **0.602** | **0.701** | **0.246** |
>
> Compared to various attention-based baselines, JoPA achieves the best performance across all metrics on the Llama-2 (7B-Chat) model.
>
> >W2. Highlighting the novelty of the proposed method.
>
> We believe **the primary novelty of our work lies in our formulation and framework for explaining the entire generation of LLMs through the joint attribution of the input prompts**. This is a largely under-explored area, as discussed in Line 038-061: while existing methods either focus on classification/next-token prediction tasks or overlook the joint effects of input attributions on the output generations, our approach aims to quantify the importance of tokens while considering their semantic interactions. To achieve this goal, we formulate and address this novel problem of measuring the combinatorial effects of tokens through a combinatorial optimization algorithm: the proposed algorithm follows a certain general principles and adapts for this unique problem of joint attribution. Additionally, our work demonstrates significant computational efficiency compared to the baseline method, Captum, with JoPA being approximately 100 times faster. The effectiveness and efficiency of our method are demonstrated through experimental results in Table 1 and Table 2.

---

> > ### Author Response · Authors · 2024-11-21
> > **Reponse to Reviewer 4Cfs -- Part 2**
> >
> > >W3. Informative case study in Sec 5.5; some downstream applications
> >
> > We will add the following discussions about the downstream applications in our Appendix.
> >
> > **Informative case study:** The case study in Section 5.5 highlights the importance of considering semantic interactions when explaining the relationship between input prompts and outputs, emphasizing the necessity of JoPA. It examines a failure case of Captum, where important tokens appear multiple times in different positions within the input prompt. Dropping a single important token does not significantly affect the semantics of the input, and consequently, the output generation remains largely unchanged. However, when multiple tokens are dropped simultaneously using JoPA, the output changes significantly compared to the original. This comparison underscores the necessity and rationale of joint attribution. Some other similar case studies are in Appendix A.2.
> >
> > **Downstream applications:** We have discussed the benefits of JoPA on improving the model safety and model diagnosis in Appendix A.3. These case study shows that JoPA could be potentially used as the defense of the jailbreak attacks and help users to design better prompts to get satisfying responses. We also do the experiments to show the effectiveness of applying JoPA to detect malicious prompts on Llama-2 (7B-Chat) model [1], which is similar to the application done in the CONTEXTCITE [8]. The results are shown below:
> >
> >
> > |      |Detection Accuracy by JoPA|ASR|
> > |:--------:|:-----:|:-----:|
> > |GCG|100$\%$|3$\%$|
> > |Prompt with Random Search|91$\%$|90$\%$|
> >
> >
> > The table presents the detection accuracy of JoPA against different adversarial prompts generated by GCG attacks [5] and Prompt with Random Search [6]. For Prompt with Random Search, the ASR is 90%, while JoPA achieves a detection accuracy of 91%. This indicates that JoPA successfully identifies 91% of malicious prompts, highlighting its potential utility as a defense mechanism.

---

> ### Comment · Reviewer_4Cfs · 2024-11-29
> **Update**
>
> Yes I have already read the response and increased my score to 6 since the response addressed my major concern on missing baseline comparisons.

---

### Official Review · Reviewer_YhkJ · 2024-11-04

**Soundness:** 3
**Presentation:** 3
**Contribution:** 3
**Rating:** 6
**Confidence:** 4

**Summary:**

This paper pays attention to the contribution of input prompts on generated content. By using counterfactual explanation, this paper proposes JoPA, a framework to highlight which components of input prompts have the fundamental  effect on the generated context via solving a combinatorial optimization problem. JoPA uses the gradient information and the probabilistic search-space strategy to find the more important tokens and improve efficiency.

**Strengths:**

1. This paper focuses on an important area, the impact of prompt on generated outputs.

2. The time cost of JoPA is quite less than baselines.

**Weaknesses:**

1. The counterfactual explanation is emphasized as the important part in JoPA, but its concept and logic are not fully explained. And there are not enough case studies to support this argument.

2. The practical usage scene of JoPA is under discussion. How would a user perform JoPA when using LLM via web service or APIs like chatgpt? Can more general advices are given for users?

**Questions:**

See weaknesses.

---

> ### Author Response · Authors · 2024-11-21
> **Reponse to Reviewer YhkJ -- Part 1**
>
> We thank the reviewer for the insightful suggestions.
>
> >W1. Concept and logic of the counterfactual explanation not being fully explained. Need enough case studies to support this argument.
>
> **Counterfactual Explanation:** We apologize for any confusions. Our work JoPA is to explain the generation of LLMs by identifying a small changes (i.e., masking few tokens) needed to alter the outcome (i.e., minimizing the probability of the original generation). This can be understood as counterfactual explanation, since it answers a "what if" question: if these input tokens are masked, the LLM would not generate the response. Namely, by applying the mask $m$ on the input prompt $x$, the generation probability of its original response $y$ would be largely decreased (i.e., $p_{\theta}(y|m\odot x) \ll p_{\theta}(y|x)$). This suggests that the newly generated output $y'$ resulting from the masked prompt would differ substantially from the original outcome $y$, as measured by our metrics in Table 1: the semantic similarity (SentenceBert) and word similarity (BLEU and ROUGE-L) between $y$ and $y'$ are much smaller than those from baselines.
>
> **Case Study:** Thank you for the suggestion, and we have included the following case studies to show the outcome changes. For instance, given the input prompt to the model:
> ```
> Follow the law of supply and demand, describe what would happen to the price of a good if the demand increased.
> ```
> The model would generate the response: "\nIn economics, the law of supply and demand states that the price of a good is determined by the interaction between the quantity of the good that...". After masking the 3 explanatory tokens: 'supply', 'price', and 'demand', the generated response would change significantly: "However, I must inform you that the question you've provided is not factually coherent, and I cannot provide an answer that may not be accurate or safe...".
>
> Similarly, given the input prompt for the model to fill in the blank:
> ```
> Fill in the blank with a word or phrase The most successful team in the NBA is the ____
> ```
> the original response is: "The most successful team in the NBA is the Golden State Warriors". If we masked out the explanatory tokens 'most', 'successful', and 'NBA' from the input, the output would change to: "\nBased on the context you provided, the word that best fits the blank is \"best.\" So, the sentence would read:\nThe team in the office is the best."
>
> The case studies of diverse tasks above demonstrate that the new responses generated from the masked inputs differ significantly from the original outputs, which highlights the impact of these explanatory tokens in influencing and altering the model's output.

---

> > ### Author Response · Authors · 2024-11-21
> > **Reponse to Reviewer YhkJ -- Part 2**
> >
> > >W2. Discussion on pratical usage. Performance of JoPA when using LLM via web service or APIs like chatgpt. More general advices given for users?
> >
> > **Pratical Usage:** Our method has potential benefits on imporving the model safety and helping in the model diagnosis as discussed in Appendix A.3. By applying our method to detect malicious tokens in the input prompt for jailbreak attacks, the model can refuse to respond to malicious queries, thereby making the attacks ineffective. Moreover, by identifying the important input attributions, the relevance between the input prompt and output generations will be assessed, contributing to the model diagnosis. We design an experiment, where we apply our method as a defense to detect the jailbreak prompts constructed by the attack method GCG [5] and Prompt with Random Search [6] respectively. The results are shown below:
> >
> > |      |Detection Accuracy by JoPA|ASR|
> > |:--------:|:-----:|:-----:|
> > |GCG|100$\%$|3$\%$|
> > |Prompt with Random Search|91$\%$|90$\%$|
> >
> > The table presents the detection accuracy of JoPA against these two jailbreaking attacks on Llama2-7b-chat-hf. For Prompt with Random Search, while its attack success rate (ASR) is 90% [1], JoPA achieves a detection accuracy of 91%. This indicates that JoPA successfully identifies 91% of malicious prompts, indicating its potential utility as a defense mechanism.
> >
> > **Perform Interpretation on Web/API-based LLMs:** The input attribution interpretation methods (ours and baselines) mainly focuses on explaining the generation of open-source LLMs, as it requires access to mask gradients for initialization and model logits for optimizing the token masks. Without access to the internal workings of a model, it is challenging to accurately quantify the importance of token attributions. While this line of interpretation works is not designed for black-box models, we believe it can be integrated to online services by the developers to enhance users' trust to these models.
> >
> > **General Advices for Users:** Our method can assist users in refining their prompts and suggest better prompt designs. As discussed in Appendix A.3, model providers can leverage our approach to highlight important tokens in the input, helping users identify requests that may have been overlooked by the model. This allows users to revise and regenerate their prompts, enabling them to obtain more relevant and satisfying responses.

---

> > > ### Comment · Reviewer_YhkJ · 2024-12-03
> > > **Acknowledgement**
> > >
> > > Thanks for the response and revision, which have addressed my doubt about the concept and logic counterfactual explanation. The authors have also clarified the practical usage. I do not have further questions.

---

### Author Response · Authors · 2024-11-21
**General Response to All Reviewers**

We thank all the reviewers for the constructive and insightful comments to help clarify and strengthen our work. We are encouraged to find that the reviewers appreiciate:

1) the importance (YhkJ) and realness (AFbs) of the proposed question of explaining the joint effect of the prompt on the output; 2) low time cost (YhkJ), efficiency (ZoiX) and advantages (AFbs) of our proposed method; 3) reasonable experiment setup (4Cfs), and comprehensive (4Cfs, ZoiX) and robust (ZoiX) of the evaluation metrics; 4) positive results (4Cfs); and 5) well-organized (4Cfs) and good-writing (ZoiX) of our paper.


**We have conducted new experiments suggested by the reviewers:**
1. We evaluated other baselines including attention-based method and ReAGent to verify the effectiveness of JoPA.
2. We provided an analysis of JoPA's ability to detect malicious prompts as a potential downstream application.
3. We conducted experiments with a larger dataset to further demonstrate the effectiveness of our method.

**We have clarified all questions raised by the reviewers:**
1. We explicitly clarified the concept and logic behind our counterfactual explanation method and included case studies for better understanding.
2. We provided illustrations and examples of practical applications of our method in real-world scenarios, such as improving model safety and enhancing prompt design.
3. We emphasized our contribution in proposing a general interpretation framework for explaining LLMs' entire generation by jointly attributing input prompt tokens with the consideration of the semantic interations in the input.
4. We conducted a detailed analysis of our algorithm's design, including mask initialization with only partial tokens and convergence, demonstrating its effectiveness.

We kindly request that you inform us of any remaining concerns or ambiguities. We are more than willing to address additonal questions and conduct further experiments should the reviewers deem it necessary.

**References in our rebuttal:**

[1] Chao, P., Debenedetti, E., Robey, A., Andriushchenko, M., Croce, F., Sehwag, V., Dobriban, E., Flammarion, N., Pappas, G.J., Tramèr, F.S., Hassani, H., & Wong, E. (2024). JailbreakBench: An Open Robustness Benchmark for Jailbreaking Large Language Models. ArXiv, abs/2404.01318.

[2] Chevalier, A., Wettig, A., Ajith, A., & Chen, D. (2023). Adapting Language Models to Compress Contexts. ArXiv, abs/2305.14788.

[3] Liu, J., Li, L., Xiang, T., Wang, B., & Qian, Y. (2023). TCRA-LLM: Token Compression Retrieval Augmented Large Language Model for Inference Cost Reduction. Conference on Empirical Methods in Natural Language Processing.

[4] Haiyan Zhao, Hanjie Chen, Fan Yang, Ninghao Liu, Huiqi Deng, Hengyi Cai, Shuaiqiang Wang, Dawei Yin, and Mengnan Du. 2024. Explainability for Large Language Models: A Survey. ACM Trans. Intell. Syst. Technol. 15, 2, Article 20 (April 2024), 38 pages.

[5] Zou, A., Wang, Z., Carlini, N., Nasr, M., Kolter, J. Z., & Fredrikson, M. (2023). Universal and transferable adversarial attacks on aligned language models. arXiv preprint arXiv:2307.15043.

[6] Andriushchenko, M., Croce, F., & Flammarion, N. (2024). Jailbreaking leading safety-aligned llms with simple adaptive attacks. arXiv preprint arXiv:2404.02151.

[7] Miglani, V., Yang, A., Markosyan, A. H., Garcia-Olano, D., & Kokhlikyan, N. (2023). Using captum to explain generative language models. arXiv preprint arXiv:2312.05491.

[8] Cohen-Wang, B., Shah, H., Georgiev, K., & Madry, A. (2024). Contextcite: Attributing model generation to context. arXiv preprint arXiv:2409.00729]

[9] Xu, Z., Jain, S., & Kankanhalli, M. (2024). Hallucination is inevitable: An innate limitation of large language models. arXiv preprint arXiv:2401.11817.

---

### Note · Authors · 2024-12-16

I have read and agree with the venue's withdrawal policy on behalf of myself and my co-authors.